# High Resolution 3D Winds Derived from a Modified WISSDOM Synthesis Scheme using Multiple Doppler Lidars and Observations

**Chia-Lun Tsai[1,3], Kwonil Kim[1], Yu-Chieng Liou[2], and GyuWon Lee*[1]**

[1]Department of Astronomy and Atmospheric Sciences, Center for Atmospheric REmote sensing (CARE), Kyungpook National University, Daegu, South Korea

[2]Department of Atmospheric Sciences, National Central University, Jhongli, Taiwan

[3]Department of Atmospheric Sciences, Chinese Culture University, Taipei, Taiwan

* Corresponding author: Prof. GyuWon Lee, E-mail: gyuwon@knu.ac.kr

**Abstract**
The WISSDOM (Wind Synthesis System using Doppler Measurements) synthesis scheme
was developed to derive high-resolution 3-dimensional (3D) winds under clear-air conditions.
From this variational-based scheme, detailed wind information was obtained from scanning
Doppler lidars, automatic weather stations (AWS), sounding observations, and local reanalysis
datasets (LDAPS, Local Data Assimilation and Prediction System), which were utilized as
constraints to minimize the cost function. The objective of this study is to evaluate the
performance and accuracy of derived 3D winds from this modified scheme. A strong wind event
was selected to demonstrate its performance over complex terrain in Pyeongchang, South Korea.
The size of the test domain is $12 \times 12$ km$^2$ extended up to 3 km height mean sea level (MSL) with
remarkably high horizontal and vertical resolution of 50 m. The derived winds reveal that
reasonable patterns were explored from a control run, as they have high similarity with the
sounding observations. The results of intercomparisons show that the correlation coefficients
between derived horizontal winds and sounding observations are 0.97 and 0.87 for u- and v-
component winds, respectively, and the averaged bias (root mean square deviation, RMSD) of
horizontal winds is between −0.78 and 0.09 (1.77 and 1.65) m s$^{-1}$. The correlation coefficients
between WISSDOM-derived winds and lidar QVP (quasi-vertical profile) are 0.84 and 0.35 for
u- and v-component winds, respectively, and the averaged bias (RMSD) of horizontal winds is
between 2.83 and 2.26 (3.69 and 2.92) m s$^{-1}$. The statistical errors also reveal a satisfying
performance of the retrieved 3D winds; the median values of wind directions are −5~5 (0~2.5)
degrees, the wind speed is approximately −1~3 m s$^{-1}$ (−1~0.5 m s$^{-1}$) and the vertical velocity is
–0.2~0.6 m s$^{-1}$ compared with the lidar QVP (sounding observations). A series of sensitivity tests
with different weighting coefficients, radius of influence (RI) in interpolation and various
combination of different datasets were also performed. The results indicate that the present setting
of the control run is the optimal reference to WISSDOM synthesis in this event and will help
verify the impacts against various scenarios and observational references in this area.

## 1. Introduction

In the past few decades, many practical methods have been developed to derive wind information by using meteorological radar data (Mohr and Miller, 1983, Lee et al., 1994, Liou and Chang, 2009, Bell et al. 2012). The derived winds substantially revealed reasonable patterns compared with conventional observations (such as surface stations, soundings, wind profiles, etc.) and models (Liou et al., 2014, North et al., 2017, Chen, 2019, Oue et al., 2019). Most comprehensive applications of the derived winds were adopted to document kinematic and precipitation structures associated with various weather systems or phenomena at different scales from thousands [cold fronts and low-pressure systems (LPS)] over hundreds (tropical cyclones and typhoons) to a couple of kilometers [convective lines and tropical cyclone rainbands (it naturally depends on the length and width of the rainbands)] (Yu and Bond, 2002, Yu and Jou, 2005, Yu and Tsai, 2013, Yu and Tsai, 2017, Tsai et al. 2018, Yu et al., 2020, Cha and Bell, 2021, Tsai et al., 2022). In addition, the accuracy of 3D winds could be improved when increasing the numbers of Doppler radar because relatively fewer assumptions and more information can be included (Yu and Tsai 2010, Liou and Chang, 2009). Therefore, the retrieved schemes within multiple Doppler radars are a more popular way to obtain high-quality 3D winds and have been extensively applied to meteorological analyses.

The technique of velocity track display (VTD, Lee et al., 1994) and ground-based velocity track display (GBVTD, Lee et al., 1999) can derive the winds from single Doppler radar under some assumptions, as the wind patterns are generally uniform or axisymmetric rotational (Cha and Bell, 2021). More extended techniques based on VTD and GBVTD have also been applied to increase the quality of derived wind data, and such techniques include Extended-GBVTD (EGBVTD, Liou et al., 2006) and generalized velocity track display (GVTD, Jou et al., 2008). However, winds usually present nonuniform patterns and fast-evolving characteristics in most mesoscale weather systems and microscale phenomena, and complete and detailed winds are still difficult to resolve by these techniques. Most developed techniques are based on the contexts of

weaknesses from the above schemes on wind retrievals. Instead of a single Doppler radar,
multiple Doppler can retrieve better quality 3D winds with relativity fewer assumptions because
they provide sufficient radial velocity measurements and wind information with wider coverage
in the synthesis domain.
Cartesian Space Editing, Synthesis, and Display of Radar Fields under Interactive Control
(CEDRIC, Mohr and Miller, 1983) is a traditional package used to retrieve 3D winds by dual-
Doppler radar observations. This scheme usually determines the horizontal winds by using two
radars, and the vertical velocity can be obtained by variational adjustment with anelastic
continuity equation. Spline Analysis at Mesoscale Utilizing Radar and Aircraft Instrumentation
(SAMURAI) software is another way to retrieve 3D winds (Bell et al., 2012); this scheme is a
kind of variational data assimilation that adopts multiple radars. Recently, Tsai et al. (2018)
utilized the measurements of six Doppler radars to document precipitation and airflow structures
over complex terrain on the northeastern coast of South Korea via Wind Synthesis System using
Doppler Measurements [WISSDOM, Liou and Chang (2009)]. The scientific studies and
applications of WISSDOM were well documented in Liou et al. (2012) and Liou et al. (2016). In
addition, Immersed Boundary Method (IBM, Tseng and Ferziger, 2003) was applied in
WISSDOM. Since one of the advantages of WISSDOM is that it considers the orographic forcing
on Cartesian coordinates by applying the IBM, higher quality 3D winds can be derived well over
terrain (Liou et al., 2013, 2014, Lee et al., 2018).
Generally, radial velocity is measured by detecting the movement of precipitation particles
relative to the locations of Doppler radars; thus, there are no sufficient radial velocity
measurements under clear-air conditions. However, the winds in clear-air conditions usually play
an important role in the initiations of various weather systems and phenomena, such as downslope
winds, gap winds, and wildfires (Reed, 1931, Colle and Mass, 2000, Mass and Ovens, 2019, Lee
et al., 2020). Although surface stations, soundings, and wind profilers can measure winds under
clear-air conditions, relatively poor spatial coverage is still a problem for obtaining sufficient

wind information in certain local areas. Therefore, scanning Doppler lidars will be one approach

to obtain wind information under clear-air conditions. Päschke et al. (2015) assessed the quality

of wind derived by Doppler lidar with a wind profiler in a year trial, and the results showed good

agreement in wind speed (the error ranged between 0.5 and 0.7 m s$^{-1}$) and wind direction (the

error ranged between 5° and 10°). Bell et al. (2020) combined an intersecting range height

indicator (RHI) of six Doppler lidars to build "virtual towers" (such as wind profilers) to

investigate the airflow over complex terrain during the Perdigäo experiment. These virtual towers

can fill the gap in wind measurements above meteorological towers. The uncertainty of wind

fields is also reduced by adopting multiple Doppler lidars (Choukulkar et al., 2017), and a high

spatiotemporal resolution of derived wind is allowed to check small-scale rotors in mountainous

areas (Hill et al., 2010).

The original WISSDOM was designed to retrieve 3D winds based on Doppler radar

observations and background inputs combined with conventional observations and modeling.

However, the original WISSDOM only provided 3D winds under precipitation conditions. It does

not work well under clear-air conditions because Doppler radar cannot easily detect radial

velocity without precipitation particles. To obtain high-quality 3D winds under clear-air

conditions, the radial velocity observed from the scanning Doppler lidars can be used in the

modified WISSDOM. The results will allow us to investigate the initiations of precipitation

systems in advance of rainfall and snowfall, which is an essential benefit over Doppler radar data.

Furthermore, the conventional observations and modeling datasets were used as isolated

constraints in the modified WISSDOM synthesis scheme. One of the benefits of the isolated

constraints is that it is easy to synthesize any kind of wind information obtained from available

datasets and give suitable weighting coefficients with different constraints when they are

processing the minimization in the cost function. Thus, more reliable 3D winds in clear-air

conditions were well derived from this modified WISSDOM synthesis scheme.

The objective of this study is to modify the WISSDOM synthesis scheme based on the

original version to be a more flexible and useful scheme by adding any number of Doppler lidars
and conventional observations as well as modeling datasets. This modified WISSDOM will allow
us to obtain an exceedingly high spatial resolution of 3D winds (50 m was set in this study) under
clear-air conditions. A resolution of 50 m was chosen in this study, as the Doppler lidars'
respective horizontal resolution averages 40-60 m. A variety of adequate datasets were collected
during a strong wind event in the winter season during an intensive field experiment ICE-POP
2018 (International Collaborative Experiments for Pyeongchang 2018 Olympic and Paralympic
winter games). In summary, the main goal of this study is to use Doppler lidar observations to
retrieve high-resolution 3D winds over terrain with clear-air conditions via WISSDOM. In this
study, detailed principles of the modified WISSDOM and data implementation are elucidated in
the following sections. In addition, the modified WISSDOM was used to retrieve 3D winds over
complex terrain under clear-air conditions in a strong wind event. The reliability of the derived
3D winds was also evaluated and discussed with respect to conventional observations.
**2. Methodology**
**2.1 Original version of WISSDOM (WInd Synthesis System using DOppler Measurements)**
WISSDOM is a mathematically variational-based scheme to minimize the cost function, and
various wind-related observations can be used as one of the constraints in the cost function. The
3D winds were derived by variationally adjusted solutions to satisfy the constraints in the cost
function; thus, this is a gradient decent technique to converge toward a solution. The original
version of WISSDOM utilized five constraints, including radar observations (i.e., reflectivity and
radial velocity), background (combined with automatic weather stations, sounding, model or
reanalysis data), continuity equation, vorticity equation, and Laplacian smoothing (Liou and
Chang 2009). Liou et al. (2012) applied the IBM in WISSDOM to consider the topographic effect
on the nonflat surfaces. One of the advantages of IBM is providing realistic topographic forcing
without changing the Cartesian coordinate system into a terrain-following coordinate system.
More scientific documentation associated with the interactions between terrain, precipitation, and
winds in different areas can be found in Liou et al. (2016) for Taiwan and in Tsai et al. (2018) for
South Korea. The cost function can be expressed as
$$J = \sum_{M=1}^{5} J_M,\tag{1}$$

where $J_M$ is the different constraints. $J_1$ is the constraint related to the geometric relation
between radar radial Doppler velocity observations $(V_r)$ and derived one from true winds $(\mathbf{V}_t =$
$u_t\mathbf{i} + v_t\mathbf{j} + w_t\mathbf{k})$ in Cartesian coordinates [eq. (2)]. Note that $\mathbf{V}_t$ is first estimated based on the
background of the sounding observations used in this study. In the absence of background
observations, the first guess of $\mathbf{V}_t$ is set to 0.
$$J_1 = \sum_{t=1}^{2} \sum_{x,y,z} \sum_{i=1}^{N} \alpha_{1,i} \left(T_{1,i,t}\right)^2.\tag{2}$$

Since WISSDOM is a scheme that uses the 4DVAR approach, the variations between different
time steps $(t)$ should be considered, and two time steps of radar observations were collected in
this constraint and all following constraints. The $x, y, z$ indicates the locations of a given grid
point in the synthesis domain, and $i$ could be any number $(N)$ of radars (at least 1). The $\alpha_1$ is
the weighting coefficient of $J_1$ $(\alpha_2$ is the weighting coefficient of $J_2$ and so on). $T_{1,i,t}$ in eq.
(2) is defined as eq. (3):
$$T_{1,i,t} = (V_r)_{i,t} - \frac{\left(x - P_x^i\right)}{r_i}u_t - \frac{\left(y - P_y^i\right)}{r_i}v_t - \frac{\left(z - P_z^i\right)}{r_i}\left(w_t - W_{T,t}\right),\tag{3}$$

$(V_r)_{i,t}$ is the radial velocity observed by the radar $(i)$ at time step $(t)$, $P_x^i, P_y^i$ and $P_z^i$ depict the
coordinate of radar $i$. The $u_t, v_t$ and $w_t$ $(W_{T,t})$ denote the 3D winds (terminal velocity of
precipitation particles) at given grid points at the time step $t$ ; and $r_i =$
$\sqrt{(x - P_x^i)^2 + (y - P_y^i)^2 + (z - P_z^i)^2}.$
The second constraint is the difference between the background ($\mathbf{V}_{B,t}$) and true (derived)
wind field ($\mathbf{V}_t = u_t\mathbf{i} + v_t\mathbf{j} + w_t\mathbf{k}$), which is defined as

$$J_2 = \sum_{t=1}^{2} \sum_{x,y,z} \alpha_2 \big(\mathbf{V}_t - \mathbf{V}_{B,t}\big)^2. \tag{4}$$

There were several options to obtain background in the original version of WISSDOM. The most
popular background resource involves using sounding observations; however, it can only provide
homogeneous wind information for each level in WISSDOM with relatively coarse temporal
resolution (3- to 12-hour intervals). The other option is combining sounding observations with
AWS (automatic weather station) observations. Although the AWS provided wind information
with better temporal resolution (1-min), the data were only observed at the surface layer with
semirandom distributions. The last option is to combine sounding, AWS, modeling or reanalysis
datasets. However, various datasets with different spatiotemporal resolutions are not favorable
for appropriate interpolation of given grid points of WISSDOM synthesis, and the accuracy and
reliability of the background may have been significantly affected by such a variety of datasets.
Thus, these different observed or model data should be treated differently to minimize
uncertainties and improve accuracy. Therefore, one of the improvements in the modified
WISSDOM is that these inputs were individually separated into independent constraints with
flexible interpolation methods. In addition, individual constraints were calculated in two time
steps if the temporal resolution of the inputs was high enough. The sounding observations are
still a necessary dataset because the air density and temperature profile were used to identify the
height of the melting level. In this study, sounding winds were adopted to represent the
background for each level and a constraint at the same time; nevertheless, the AWS and reanalysis
dataset are independent constraints in the modified WISSDOM (details are provided in the
following section).
The third, fourth and fifth constraints in the cost function are the anelastic continuity
equation, vertical vorticity equation and Laplacian smoothing filter, respectively. Equations (5),
(6) and (7) are denoted as follows:

$$J_3 = \sum_{t=1}^{2} \sum_{x,y,z} \alpha_3 \left[ \frac{\partial(\rho_0 u_t)}{\partial x} + \frac{\partial(\rho_0 v_t)}{\partial y} + \frac{\partial(\rho_0 w_t)}{\partial z} \right]^2, \qquad (5)$$


$$J_4 = \sum_{x,y,z} \alpha_4 \left\{ \frac{\partial \zeta}{\partial t} + \overline{\left[ u\frac{\partial \zeta}{\partial x} + v\frac{\partial \zeta}{\partial y} + w\frac{\partial \zeta}{\partial z} + (\zeta + f)\left(\frac{\partial u}{\partial x} + \frac{\partial v}{\partial y}\right) + \left(\frac{\partial w}{\partial x}\frac{\partial v}{\partial y} - \frac{\partial w}{\partial y}\frac{\partial u}{\partial z}\right) \right]} \right\}^2, \quad (6)$$


$$J_5 = \sum_{t=1}^{2} \sum_{x,y,z} \alpha_5 [\nabla^2 (u_t + v_t + w_t)]^2. \qquad (7)$$


$\rho_0$ in eq. (5) is the air density, and $\zeta = \partial v/\partial x - \partial u/\partial y$ in eq. (6). The main advantage is that
using vertical vorticity can provide further improvement in winds and thermodynamic retrievals
from a method named as Terrain-Permitting Thermodynamic Retrieval Scheme (TPTRS, Liou et
al. 2019).
**2.2 The modified WISSDOM**

In addition to the five constraints in the original version, the modified WISSDOM synthesis

scheme includes three more constraints in the cost function. Thus, the cost function in the
modified WISSDOM was written as

$$J = \sum_{M=1}^{8} J_M. \qquad (8)$$


$J_1 \sim J_5$ in (8) are the same constraints corresponding to equations (2)-(7). The main purpose

of this study is to retrieve 3D winds under clear-air conditions in which observational data are
relatively rare. Instead of the radial velocity $(V_r)_{i,t}$ observed from Doppler radars in eq. (3) in
original version of WISSDOM, the radial velocity observed from Doppler lidars was adopted in
the modified WISSDOM synthesis. In addition, if there were no precipitation particles under
clear-air conditions, the terminal velocity of precipitation particles $(W_{T,t})$ was set to zero in eq.
(3) in the modified WISSDOM. In this study the time steps in WISSDOM are set to 12 min,
corresponding to the temporal resolution of the primary input lidar data. Relatively minor changes
in environmental conditions were assumed in WISSDOM due to the limitation on the coarse
temporal resolution from specific inputs. For example, the closest time step of a sounding
observation or LDAPS dataset was chosen regarding the synthesis time, and the time constrain
was set to be the same.

The sixth constraint is the difference between the derived wind fields and the sounding

observations ($\mathbf{V}_{S,t}$), as defined in (9):

$$J_6 = \sum_{t=1}^{2} \sum_{x,y,z} \alpha_6 \left(V_t - V_{S,t}\right)^2. \tag{9}$$

The sounding data in $J_6$ were interpolated to the given grid points near its tracks bearing on the
radius influence (RI) distance (the details are provided in Section 3.2.3). The main difference
between $J_6$ and $J_2$ is that the sounding data with various wind speeds and directions were used
as an observation for given 3D locations in $J_6$ instead of the constraint of homogeneous
background winds (i.e., uniform wind speed and direction) for each level in the studied domain
in $J_2$. An additional benefit of $J_6$ is that any number of sounding observations can be efficiently
adopted in the WISSDOM synthesis domain. The seventh constraint represents the discrepancy
between the true (derived) wind fields and AWS ($\mathbf{V}_{A,t}$), as expressed in (10):

$$J_7 = \sum_{t=1}^{2} \sum_{x,y,z} \alpha_7 \left(V_t - V_{A,t}\right)^2. \tag{10}$$

Finally, the eighth constraint measures the misfit between the derived winds and the local
reanalysis dataset ($\mathbf{V}_{L,t}$), as defined in (11):

$$J_8 = \sum_{t=1}^{2} \sum_{x,y,z} \alpha_8 \left(V_t - V_{L,t}\right)^2. \tag{11}$$

In this study, various observations and reanalysis datasets were utilized as constraints in the cost
function of WISSDOM. The most important dataset is the radial velocity observed from Doppler
lidars, which can measure wind information with high spatial resolution and good coverage from
near the surface up to higher layers in the test domain. Sounding and AWS can provide horizontal
winds for background or to be included in the constraints. The local reanalysis datasets were
obtained from the 3DVAR Local Data Assimilation and Prediction System (LDAPS) data
assimilation system from the Korea Meteorological Administration (KMA). Since these datasets
have different coordinate systems and various spatiotemporal resolutions, additional procedures
are required before the synthesis. Detailed descriptions of the procedures are described in the next
section.
The high-quality synthesized 3D wind field from radar observations has been applied in
several previous studies such as those by Liou and Chang (2009), Liou et al. (2012, 2013, 2014,
2016), and Lee et al. (2017). The advantages and details of the WISSDOM can be found in Tsai
et al. (2018). Although several studies have used Doppler radar in WISSDOM, this study is the
first time to apply Doppler lidar data in WISSDOM. This modified WISSDOM synthesis scheme
has also been applied in the analysis related to the mechanisms of orographically induced strong
wind on the northeastern coast of Korea (Tsai et al., 2022). In contrast to previous studies, this
study provides clear context, detailed procedures, reliability, and the limitations of the modified
WISSDOM.
**3. Data processing with a strong wind event**
**3.1 Basic information of WISSDOM synthesis**
A small domain near the northeastern coast of South Korea was selected to derive detailed
3D winds over complex terrain (in the black box in the inset map in Fig. 1) because relatively
dense and high-quality wind observations were only collected in this region during ICE-POP
2018. The size of the WISSDOM synthesis domain is $12 \times 12$ km$^2$ (up to 3 km MSL height) in
the horizontal (vertical) direction with 50 m grid spacing. Such high spatial resolution 3D winds
were synthesized every 1 hour in this test. The output time steps are adjustable to be finer
(recommended limitation is 10 mins), but they are highly related to the temporal resolution of
various datasets and computing resources. Two scanning Doppler lidars are located near the
center of the domain: one is the equipped "WINDEX-2000" (the model's name from the
manufacturer) at the May Hills Supersite (MHS) site, and the other is the "Stream line-XR" at
the DaeGwallyeong regional Weather office (DGW) site. In addition to the operational AWS
(727 stations), additional surface observations (32 stations) are also involved in ICE-POP 2018
surrounding the MHS and DGW sites and the venues of the winter Olympic Games. The
soundings are launched at the DGW site every 3 hours during the research period. The LDAPS
also provided high spatial resolution of wind information in the test domain. The horizontal
distribution of all instruments and datasets used are shown in Fig. 1.

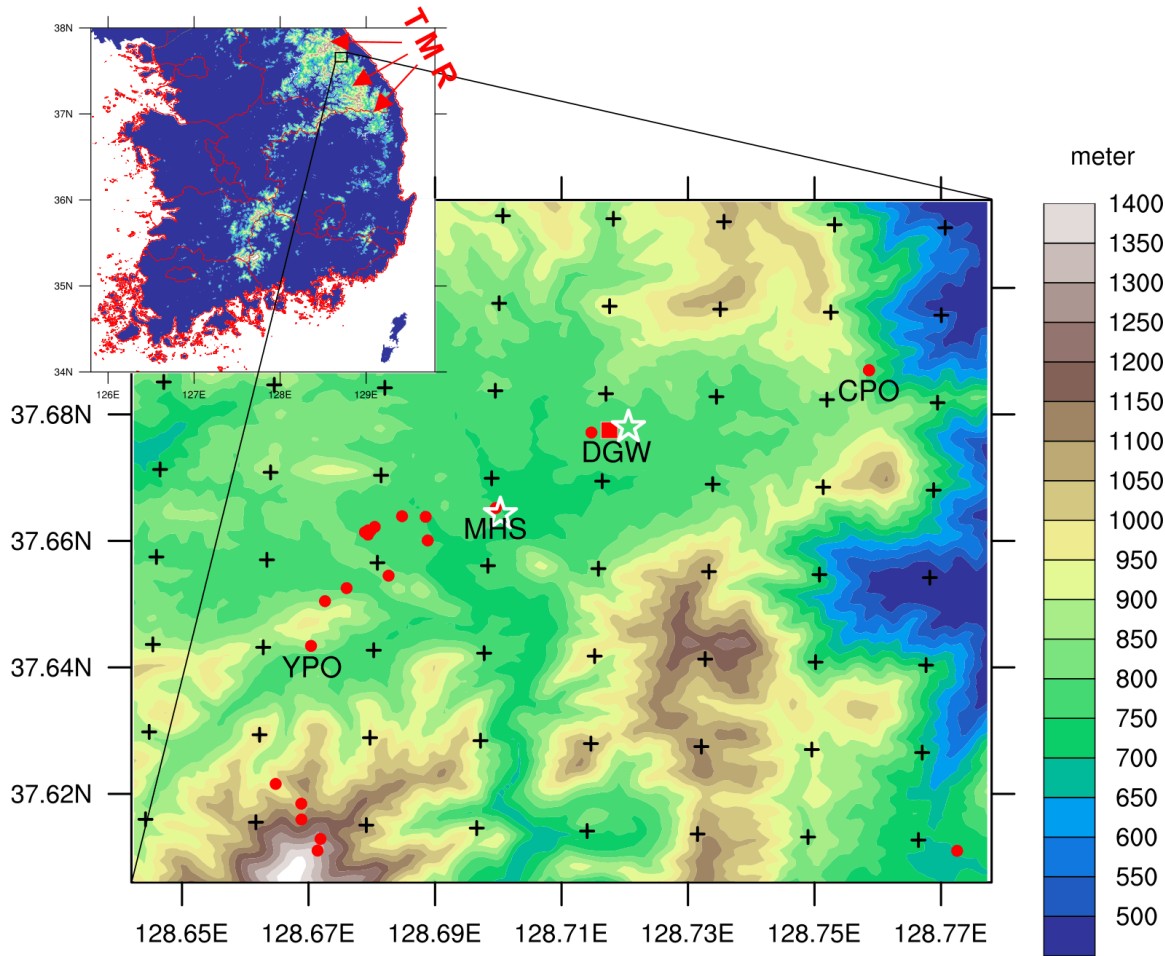

Figure 1. Horizontal distribution of instruments and datasets used in this study. A small box in the upper map

indicates the WISSDOM synthesis domain. The Doppler lidars are marked by start symbols at the MHS and

DGW sites. Red solid circles and square indicate the automatic weather station (AWS) and sounding, respectively.

The black cross marks the data points of LDAPS. Topographic features and elevations are shown with the color

shading in a color bar in the figure. The location of the Teabeak Mountain Range (TMR) is also marked.
**3.2 Data implemented in WISSDOM synthesis**
**3.2.1 Scanning Doppler lidars**
The radial velocity observed from two scanning Doppler lidars was utilized to retrieve 3D
winds via WISSDOM synthesis. The original coordinate system of observed lidar data is not a
Cartesian coordinate system but a spherical (or polar) coordinate system as a plan position
indicator (PPI) and hemispheric range height indicator (HRHI) or the RHI. Although relatively
dense and complete coverage of wind information (i.e., radial velocity of aerosols) were
sufficiently recorded by lidar observations, the collected data are usually not located directly on
the given grid points in the WISSDOM synthesis (i.e., Cartesian coordinate system). In this study,
the lidar data were interpreted simply from the lidar coordinate system to the Cartesian coordinate
system via bilinear interpolation.
The scanning strategy of the lidar at the DGW site includes five elevation angles for PPI ($7°$,
$15°$, $30°$, $45°$, and $80°$ before 10:00 UTC on 14 Feb. 2018 and $4°$, $8°$, $14°$, $25°$, and $80°$ after 10:00
UTC) and two HRHIs at azimuth angles of $51°$ and $330°$. A full volume scan included all PPIs
and HRHIs every ~12 min. The maximum observed radius distance is ~13 km, and the grid
spacing is 40 m for each gate along the lidar beam. The scanning strategy of the lidar at the MHS
site involves seven elevation angles for PPI ($5°$, $7°$, $10°$, $15°$, $30°$, $45°$, and $80°$) and one HRHI
at an azimuth angle of $0°$. A full volume scan included all PPIs and RHIs every ~12 min. The
maximum observed radius distance was ~8 km, and the grid spacing was 60 m. The vertical
distribution of lidar data in the test domain is shown as blue lines in Fig. 2a.
**3.2.2 Automatic weather station (AWS)**
Most of the AWS are not exactly located on the given grid points of the Cartesian coordinate
system. Objective analysis (Cressman, 1959) is a popular way to correct semirandom and
inhomogeneous meteorological fields into regular grid points. This study adopted objective
analysis for the AWS observations with adjustable RI distances between 100 m and 2000 m. After
this first step, the observational data can reasonably interpolate to the given grid points
horizontally. Furthermore, an additional step is required to put these interpolated data into the
given grid points at different vertical levels because the AWS are located at different elevations
in the test domain. In the traditional way of original WISSDOM, the interpolated data are moved
to the closest level with the shortest distance just above the AWS site. However, the interpolated
data are NOT moved to the closest level if the shortest distances are large like more than half
(50%) of grid spacing. Nevertheless, to include more data from the AWS observations
appropriately, adjusted distances between the AWS sites and given grid points at different vertical
levels were necessarily considered. These adjusted distances can be named as vertical extension
(VE) here, and there are two options of 50% and 90% in the tests of this study, which correspond
to 25 m and 45 m extensions between each grid (in case of the grid spacing is 50 m), respectively.
An example demonstrated how to implement the interpolated data to the given grid points by
adjustable VE after step one (Fig. 2b).

In Fig. 2b, the interpolated data do not need to move to a given grid point (as an example, at

the 800 m level here) if the elevation of the AWS is equal to the height of a given grid point as
point A. When the AWS is located higher than a given grid point (as point B in Fig. 2b) and does
not reach the lower boundary of VE (50%) from the upper given grid point (i.e., at the 850 m
level), this interpolated data will be removed and wasted. In contrast, when the interpolated data
are located just below the given grid point with 50% VE, it will be achieved in the WISSDOM
synthesis at the 800 m level (point C in Fig. 2b). The interpolated data of point D have a similar
situation to point B; however, it will be achieved at the 800 m level because a higher VE (90%)
was applied here. Since the locations of the AWS are semirandom with relatively sparse or
concentrated distributions, the optimal RI and adjustable VE make it possible to include more
AWS observations in the WISSDOM synthesis.

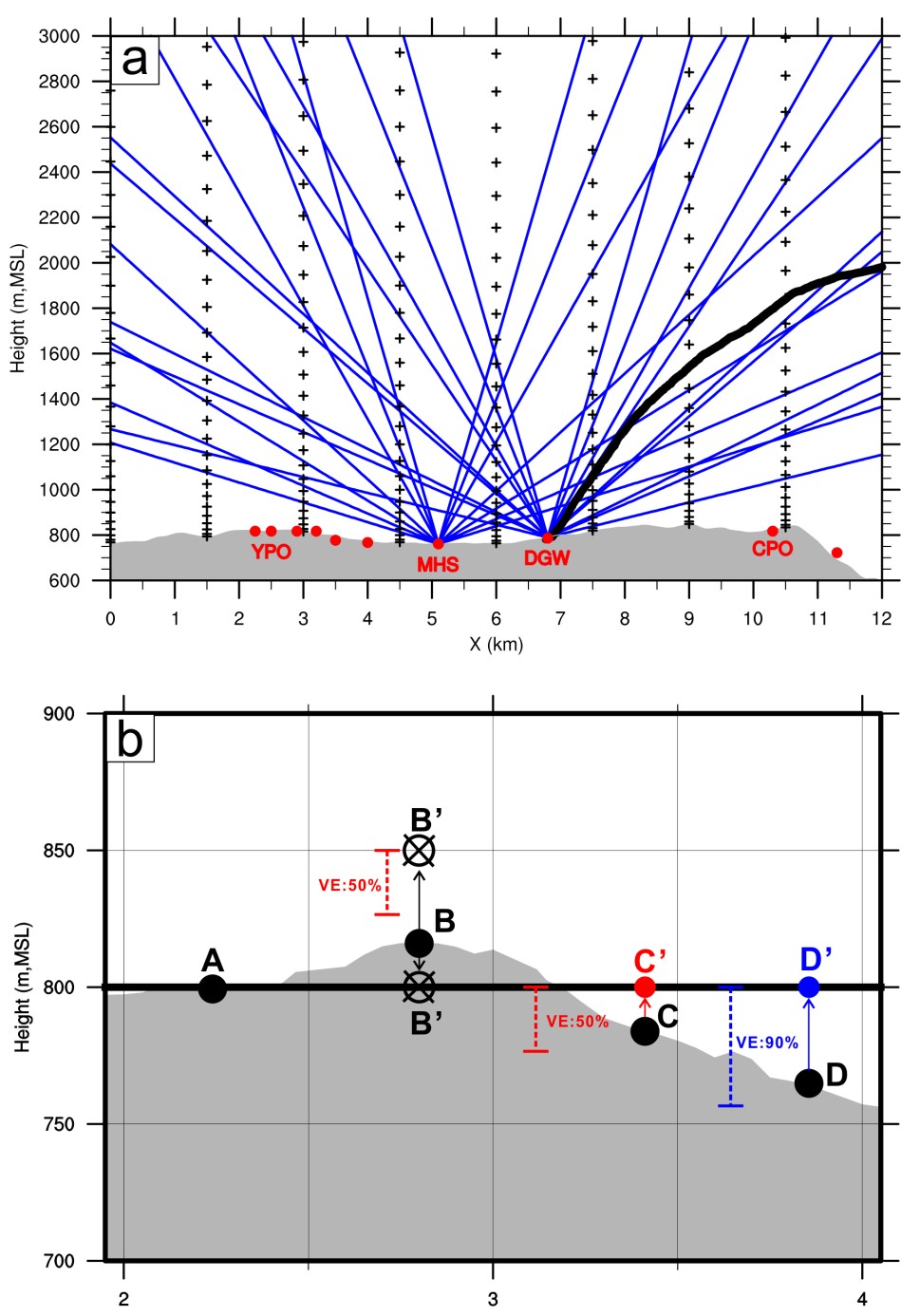

Figure 2. (a) Schematic diagram of the vertical distribution of adopted lidar datasets. Blue lines indicate the lidar
data observed at the DGW and MHS sites with different elevation angles. The AWS are located on the ground
and are marked by solid red circles. An example of a sounding track launched from the DGW site in one time
step (06:00 UTC on 14 Feb. 2018) is plotted as a thick black line. The black cross marks indicate the vertical
distribution of the LDAPS dataset. (b) Schematic diagram for data implementation with various locations of the
AWS and different percentages of VE (vertical extension) from given grid points at the 800 m MSL level (thick
black line). The gray shading on the bottom represents the topography.

### 3.2.3 Sounding

During ICE-POP 2018, the soundings are launched at the DGW site every 3 hours (from 00Z). Vertical profiles of air pressure, temperature, humidity, wind speed and directions were recorded every second (i.e., ~3 m vertical spatial resolution) associated with the rising sensor. The sounding sensor drifted when rising, and an example of its track in one time step is shown as a thick black line in Fig. 2a. In this example, the sounding movement was mostly affected by westerly winds, and it measured the meteorological parameters in any location along the track in the test domain. The coordinate system of sounding data is quite similar to the distribution of AWS measurements, and the observations are not located right on the given grid points of the WISSDOM synthesis.

Similar to the AWS data, the sounding data also underwent objective analysis with an adjustable RI distance for the wind measurements in the first step. Then, the interpolated data were switched to given grid points for each vertical level by the different VE in the WISSDOM synthesis.

### 3.2.4 Reanalysis dataset: LDAPS

The local reanalysis dataset LDAPS was generated by the KMA. This dataset provides u- and v-component winds every 3 hours, and the horizontal spatial resolution is ~1.5 km with the grid type in Lambert Conformal (as black cross marks in Fig. 1). The data revealed denser distributions near the surface and sparse distributions at higher levels (see Fig. 2a). The initiations of wind variables in the LDAPS were assimilated with many observational platforms, including radar, AWS, satellite and sounding data. Thus, the relatively high reliability of this dataset could be expected. In addition, such datasets have also significantly improved the forecast ability in small-scale weather phenomena over complex terrain in Korea (Kim et al., 2019, Choi et al., 2020, Kim et al., 2020).

The LDAPS data are not located directly on the given grid points of the WISSDOM synthesis
system. Unlike the distribution of AWS and sounding observations, LDAPS has dense and good
coverage in the test domain. The Cartesian coordinate system is the most efficient method and
the best system for partial differential equations (Armijo, 1969), and it is also used in the cost
function of WISSDOM (Liou and Chang, 2009). In this study, the horizontal and vertical
resolutions of given grid points were primarily determined by the characteristics of lidar data.
Therefore, similar to lidar observations, the LDAPS data were also interpolated to the given grid
points on the Cartesian coordinate system via the bilinear interpolation method.
**3.3 Overview of the selected strong wind event**
A strong wind event was selected to evaluate the performance of this modified WISSDOM
synthesis scheme. In this strong wind event, the evolution of surface wind patterns on the Korean
Peninsula was mainly dominated by a moving LPS which is one type of strong downslope winds
(Park et al, 2022, Tsai et al., 2022). The LPS moved out from China and penetrated the northern
part of the Korean Peninsula through the Yellow Sea beginning at approximately 12:00 UTC on
13 February 2018. Consequently, a relatively strong surface wind speed (exceeding ~17 m s$^{-1}$)
was observed when the LPS was located near the northeastern coast of the Korean Peninsula
(~130°E, 40°N) at 00:00 UTC on 14 February 2018 (Fig. 3). Then, the surface wind speed became
weak when the LPS moved away from South Korea after 00:00 UTC on 15 February 2018 (not
shown); the details of the synoptic conditions can be found in Tsai et al. (2022).

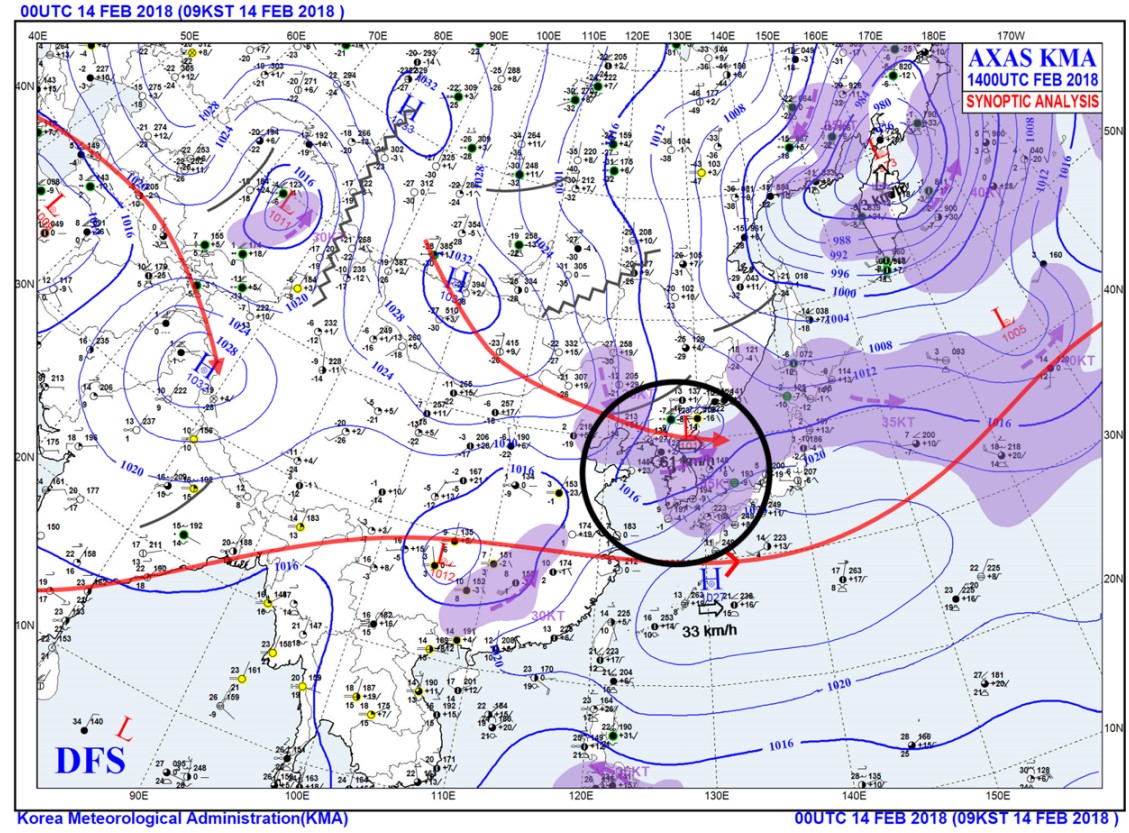

Figure 3. Synoptic surface chart from the Korea Meteorological Administration (KMA) at 00:00 UTC on 14 Feb. 2018. The locations of the Korean peninsula and the LPS has been marked by black circle.

This event is one of two strong wind events (i.e., daily maximum wind speeds larger than 10 m s$^{-1}$ observed at the AWS sites along the northeastern coast of South Korea) in the past decade based on the KMA historic record. Such a strong wind event may help us to examine the potential maximum errors of the retrieved winds. Since persistent, strong westerly winds were observed by the soundings and AWS from near the surface and upper layers over the TMR during the event, the data coverages in the test domain were checked during a chosen time step (06:00 UTC on 14 February 2018). The percentage of data occupations for each dataset (after interpolation) was checked, and the results are shown in Fig. 4. Note that the elevation of the TMR is approximately 700 m MSL in the test domain. The lidars provided good coverage of 100% to 50% at the lower layers between 700 m and 800 m MSL. The coverage of lidars was reduced significantly above 900 m MSL and remained at ~5% due to the scan strategy during the Olympic games (more dense observations near the surface). The maximum coverage of the AWS

observations is ~40% at 800 m, and there was less coverage above this layer since relatively few
AWS are located in the higher mountains. Because only one sounding observation was utilized
in this domain, relatively few coverages were also depicted. The local reanalysis LDAPS can
provide complete coverage above 900 m MSL (exceeding 100%), albeit there was less coverage
in the lower layers due to terrain. The lidar, sounding, and AWS observations covered most areas
at lower levels but not higher levels; thus, the LDAPS compensated for most of the wind
information at the upper layers in the WISSDOM synthesis.

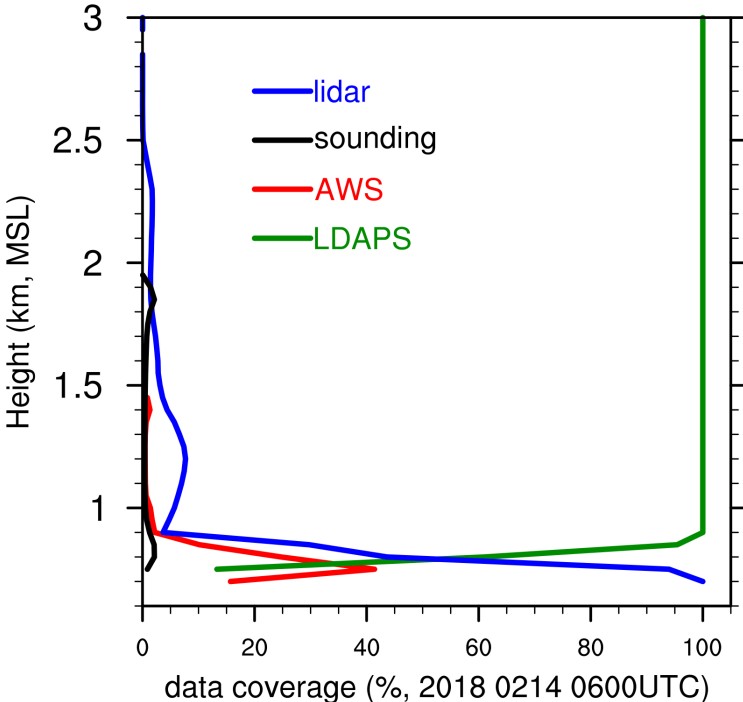


Figure 4. Data coverage (percentage, %) of the lidar (blue line), sounding (black line), AWS (red line) observations,

and LDAPS (green line) at 06:00 UTC on 14 Feb. 2018.

## 4. Control run and the accuracy of WISSDOM

### 4.1 Control run

Relatively reliable 3D winds were derived by a control run of the WISSDOM synthesis

because all available wind observations and local reanalysis datasets were appropriately acquired.
These datasets provided sufficient and complete wind information with a high percentage of
coverage in the test domain (cf. Fig. 4). Therefore, the retrieved winds from the control run can
be treated as the optimal results in WISSDOM. The control run was performed carefully with the
necessary procedures in data implementation before running the WISSDOM synthesis as follows.
The lidar and LDAPS datasets must perform bilinear interpolation to the given grid points in
WISSDOM, and the sounding and AWS observations must undergo objective analysis with the
appropriate RI distance and VE. The quantities of the weighting coefficients for each input dataset
followed the default setting from the original version of WISSDOM. The 3D winds were derived
during one time step at 06:00 UTC on 14 Feb. 2018 and compared with conventional
observations. The best weighting coefficients have been determined by a series of observation
system simulation experiment (OSSE) type tests from Liou and Chang (2009). They put more
weight on observations and less on modeling inputs. Based on the experiences and the default
setting of weighting coefficients from their studies, the basic setting of the control run was first
decided. Consequently, sensitivity tests were performed to better understand the possible
variations associated with different weighting coefficients when the lidar data were implemented.
The basic setting of this control run is summarized in Table 1.

Table 1    Basic setting of WISSDOM (control run)

| | |
|---|---|
| Domain Range | Latitude: 37.606°N~37.713°N<br>Longitude: 128.642°E~128.778°E |
| Domain Size | 12 × 12 × 3 km (long × width × vertical) |
| Spatial Resolution | 0.05 × 0.05 × 0.05 km (long × width × vertical) |
| Terrain Resolution | 0.09 km |
| Coordinate System | Cartesian coordinate system |
| Background | Sounding (DGW) |
| Data Implementation | Doppler Lidars (MHS, DGW): bilinear interpolation<br>AWS: objective analysis (RI*: 1 km, VE*: 90%)<br>Sounding (DGW): objective analysis (RI: 1 km, VE: 90%)<br>LDAPS: bilinear interpolation |
| Weighting Coefficient (input datasets) | Doppler Lidars ($\alpha_1$): $10^6$<br>Background ($\alpha_2$):$10^2$ |

Sounding ($\alpha_6$): $10^6$
AWS ($\alpha_7$): $10^6$
LDAPS ($\alpha_8$): $10^3$

*RI: radius influence, VE: vertical extension

422  The results of 3D winds at 800 m MSL derived from the control run are shown in Figs. 5a,

423 c, and e. Topographic features comprised relatively lower elevations in the center of the test

424 domain, and there were weaker u-component winds ($\sim$7 m s$^{-1}$) near the AWS and MHS lidar sites

425 between 128.67°E and 128.71°E (Fig. 5a). In contrast, the u-component winds ($\sim$15 m s$^{-1}$) were

426 almost doubled near the DGW lidar site (between 128.71°E and 128.73°E). The vertical

427 structures of the u-component winds across these two lidars (i.e., along the black line in Fig. 5a)

428 are shown in Fig. 5b. The strength of the u-component winds rapidly increased from the surface

429 to the upper layers (from $\sim$6 to 20 m s$^{-1}$), and uniform u-component winds with wavy pattern

430 were depicted above $\sim$1 km MSL except for the stronger winds near the surface surrounding the

431 DGW site. There were relatively weak (strong) u-component winds surrounding the lidar at the

432 MHS (DGW) site near the surface. Relatively weak v-component winds were found

433 (approximately ±4 m s$^{-1}$) at 800 m MSL (Fig. 5c); thus, the horizontal wind directions were

434 mostly westerly winds during this time step. The v-component winds were obviously accelerated

435 in several local areas encompassing the terrain (near 128.71°E). The vertical structure of the v-

436 component winds (Fig. 5d) indicates that the v-component winds became stronger in the upper

437 layer. The wind directions were changed from westerly to southwesterly from the near surface

438 up to $\sim$1.4 km MSL height. Updrafts were triggered on windward slopes when westerly winds

439 impinge the terrain or hills (Figs. 5e and 5f). Basically, the 3D winds derived from the WISSDOM

440 synthesis reveal reasonable patterns compared to synoptic environmental conditions (cf. Fig. 3);

441 the moving LPS accompanied stronger westerly winds.

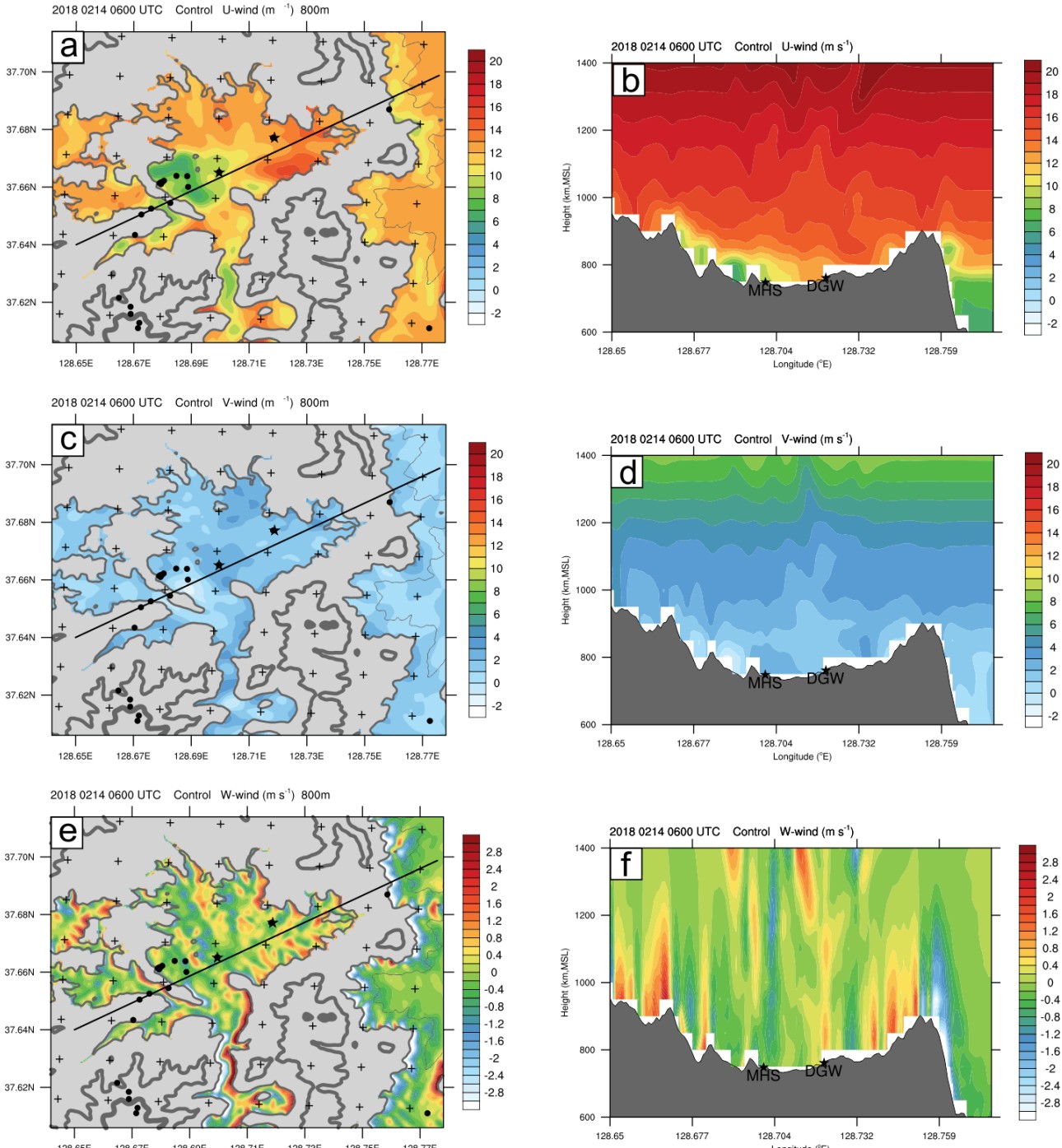

Figure 5. The 3D winds were derived from the control run by the WISSDOM synthesis at 06:00 UTC on 14 Feb.
2018. (a) The u-component winds (color, m s⁻¹) at 800 m MSL; the gray shading represents the terrain area, and
the contours indicate different terrain heights of 600 m, 800 m and 1000 m MSL corresponding to thin to thick
contours. The locations of lidars are marked with asterisks. (b) Vertical structures of u-component winds (color,
m s⁻¹) along the black line in (a) The gray shading in the lower part of the figure indicates the height of the terrain.
(c) and (d) are the same as (a) and (b) but for the v-component winds. (e) and (f) are the same as (a) and (b) but
for the w-component winds.

## 4.2 Intercomparison between derived winds and observations

Detailed analyses were performed in this section to quantitatively evaluate the accuracy of the optimally derived 3D winds from the WISSDOM synthesis. Two kinds of instruments were available in the test domain to detect the relatively realistic winds: sounding and lidar quasi-vertical profiles (QVP, Ryzhkov et al., 2016). The QVP of horizontal and vertical winds were retrieved based on the velocity-azimuth display (VAD) technique (Browning and Wexler, 1968, Gao et al., 2004). We regressed the Fourier coefficients of the Doppler velocities of the 80° PPI under the linear horizontal wind assumption and obtained the horizontal wind profile. The vertical (i.e., w-component) wind was retrieved under the assumptions of constant vertical wind, zero terminal velocity of aerosol particles, and no horizontal divergence [see Kim et al. (2022) for details on the wind retrieval]. The accuracy of the retrieved wind profile is suitable for the WISSDOM wind evaluation, given the low root mean square deviation (RMSD) of $< 2.5$ m s$^{-1}$ and high correlation coefficient of $> 0.94$ of horizontal wind speed as shown in the comparison against 487 rawinsondes (Kim et al., 2022). The horizontal winds observed from the soundings and the u-, v-, and w-component winds of the lidar QVP at the DGW site were utilized to represent the observations.

A complete analysis of the intercomparison between the WISSDOM synthesis and observations is presented in the following subsections. Because the verification observations are being used in the WISSDOM synthesis, the results of the control run are not verified independently; nevertheless, detailed discussions regarding the results of the sensitivity tests for the observations are presented in Section 5.

### 4.2.1 Sounding

The discrepancies in horizontal winds derived from WISSDOM and the sounding observations for the entire research period (from 12:00 UTC on 13 to 12:00 UTC on 14 February

2018) were analyzed. Fig. 6 shows the scatter plots of the u- and v-component winds on the
locations following the tracks of sounding launched from the DGW site. Most of the u-component
winds derived from WISSDOM are in good agreement with the sounding observations, and the
wind speed is increased with the height from approximately 10 to 40 m s$^{-1}$. Slight underestimation
of retrieved u-component winds can be found at the layers of 1.5~2 km MSL (Fig. 6a). In contrast,
most of the v-component winds were weak (smaller than 15 m s$^{-1}$) at all layers, because the
environmental winds were more like westerlies during the research period. There were also
slightly overestimated v-component winds derived from WISSDOM at the layers of 1.5~2 km
MSL (Fig. 6b). The possible reason why the overestimated winds occurred above ~1.5 km MSL
is that lidar data had relatively less coverages at higher layers (cf. Fig. 4).

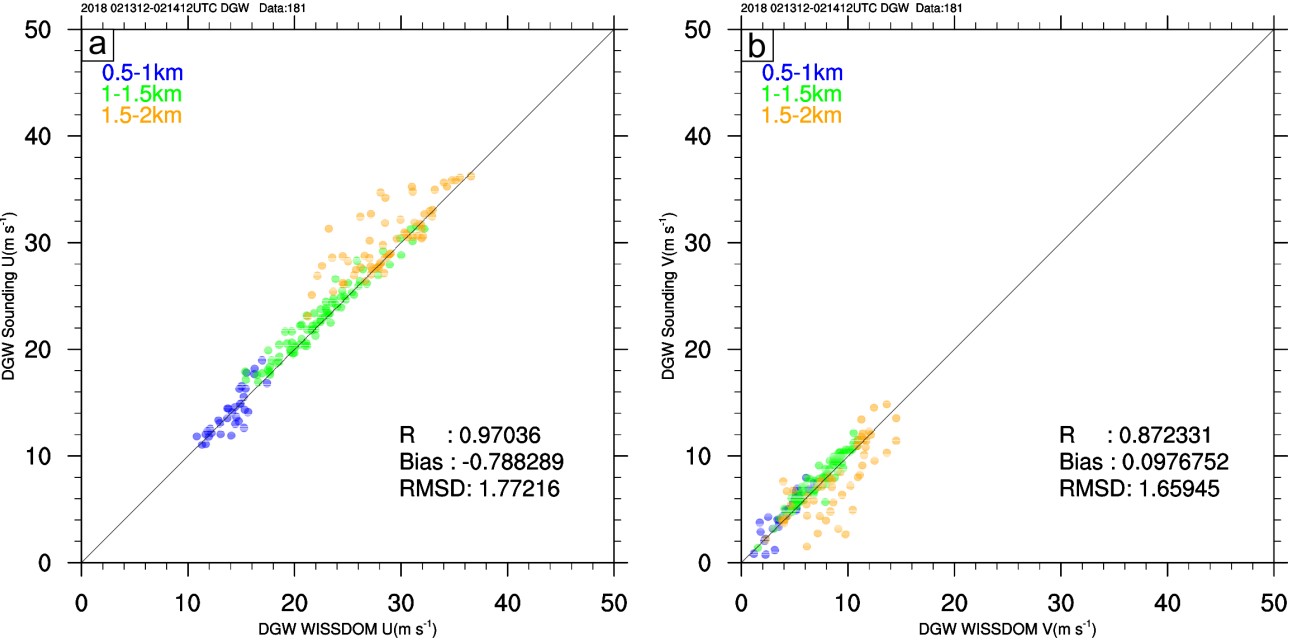

Figure. 6. Scatter plots of (a) u-component winds between the WISSDOM synthesis (x-axis) and sounding
observations (y-axis) above the DGW site during the research period. The colors indicate different layers, and
the numbers of data points, correlation coefficients, average biases and root mean square deviations are also
shown in the figure. (b) The same as (a) but for v-component winds.
Overall, the u-component winds show a high correlation coefficient (exceeding 0.97), low
average bias (−0.78 m s$^{-1}$), and the RMSD of 1.77 m s$^{-1}$. The correlation coefficient of the v-
component is also high (0.87), the average bias is 0.09 m s$^{-1}$, and the RMSD is 1.65 m s$^{-1}$.

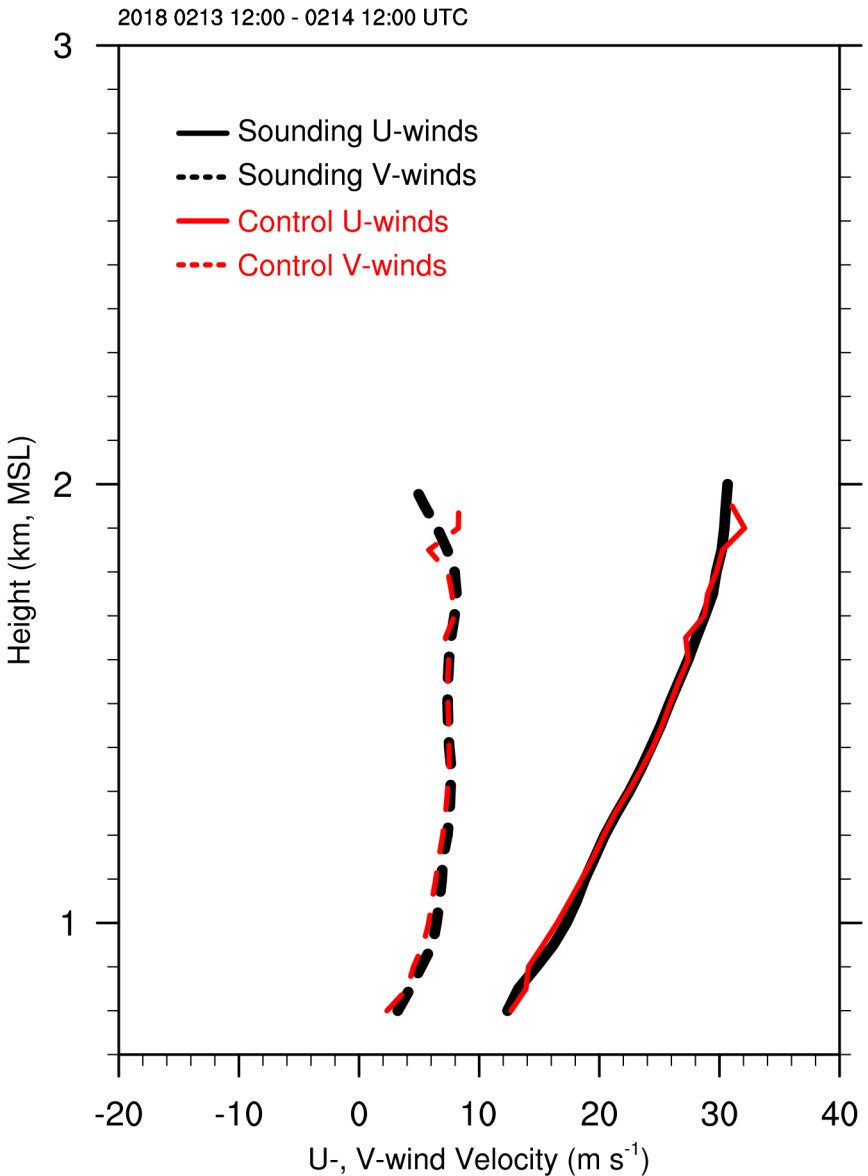


Figure 7. Vertical wind profiles of average horizontal winds derived from the WISSDOM synthesis (red lines and
vectors) and sounding observations (black lines and vectors) above the DGW site from 12:00 UTC on 13 to
12:00 UTC on 14 Feb. 2018. Solid lines indicate u-component winds (m s$^{-1}$), and dashed lines indicate v-
component winds (m s$^{-1}$).

The vertical profiles of the averaged u- and v-component winds for the period of 12:00 UTC

on 13 to 12:00 UTC on 14 Feb. 2018 is shown in Fig. 7 for the WISSDOM synthesis (red) and
sounding observations (black) launched from the DGW site. The average profiles agree well
except for the height above 1.5 km MSL, slight discrepancies of u- and v-component winds (< 1
m s$^{-1}$). Their statistical errors during the entire research period were quantified by the box plot
shown in Fig. 8.

The maximum difference in wind directions between the WISSDOM synthesis and sounding

observations is small at all layers. Only relatively larger IQR and median values can only be
found at the lowest level. The interquartile range (IQR) and median values of the wind direction
differences are smaller (between ~0 and 2.5 degrees) during the entire research period (Fig. 8a).
Basically, the IQR and median values of the wind direction differences are close to 0 degrees
above 1 km MSL. Fig. 8b shows the difference in wind speed between the WISSDOM synthesis
and sounding observations. The differences of wind speed derived from WISSDOM were slightly
underestimated in the layers between ~0.85 and 1.3 km MSL. The median values of the wind
speed differences were between $-1$ and $0.5$ m s$^{-1}$, and the IQR of wind speed differences were
between $-2$ and $0.5$ m s$^{-1}$. Above 1.3 km MSL, the differences in wind speed were small as their
median values are close to 0 m s$^{-1}$.

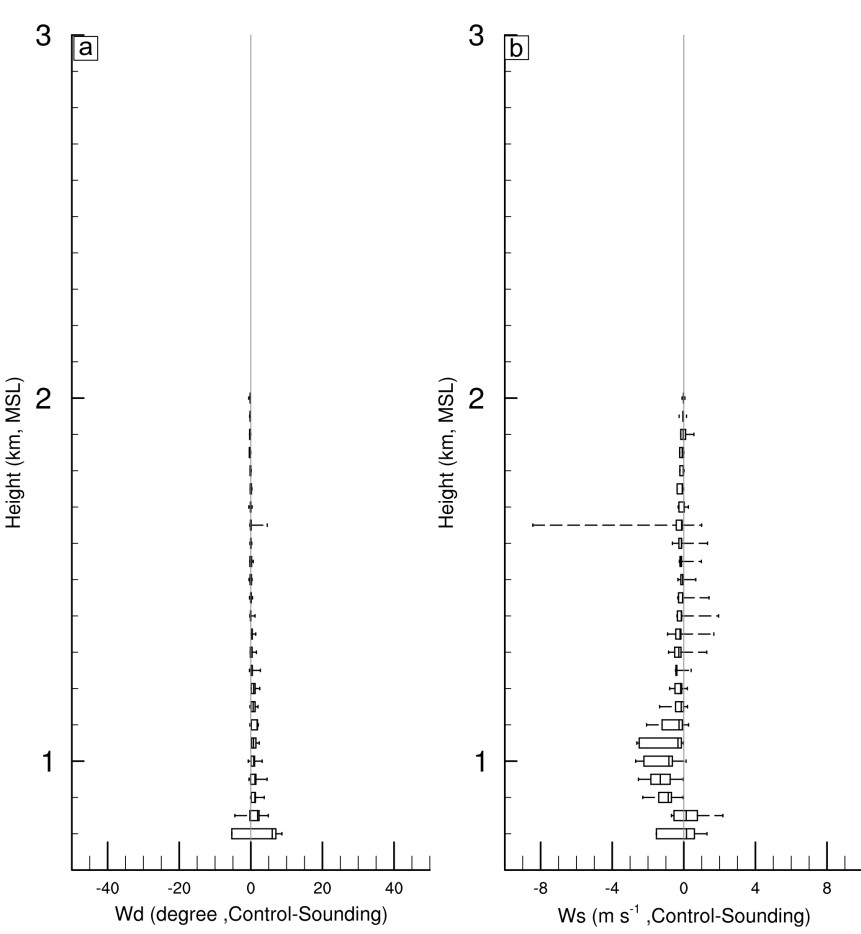


Figure 8. The box plot of average (a) wind direction discrepancies between the WISSDOM synthesis and sounding

observations above the DGW site during the research period. (b) Same as (a) but for the wind speed.

### 4.2.3 Lidar QVP

The lidar QVP is another observational reference used to evaluate the performance of derived winds from the WISSDOM synthesis. The scatter plots of the horizontal winds derived from WISSDOM and lidar QVP at the DGW site are shown in Fig. 9. The strength of the u-component winds increases with height in the range between approximately 10 m s⁻¹ and 40 m s⁻¹ from the surface up to ~2.5 km MSL (Fig. 9a). Although the results show a relatively high correlation coefficient (0.84) for the u-component winds from lower to higher layers in the entire research period, the degree of scatter is larger than that in Fig. 6a. The average bias and RMSD of the u-component winds are 2.83 m s⁻¹ and 3.69 m s⁻¹, respectively. The correlation coefficient of v-component winds is lower (0.35) in association with low wind speed (<15 m s⁻¹) from the surface to 2.5 km MSL (Fig. 9b), and it may possibly relate to less coverage from lidar QVP data at higher layers. The average bias and RMSD of the v-component winds are 2.26 m s⁻¹ and 2.92 m s⁻¹, respectively. The results of these scatter plot analyses are summarized in Table 2. Basically, the u-component winds have high correlations, relatively lower bias, and lower RMSD than the v-component winds because the environmental winds are more westerly.

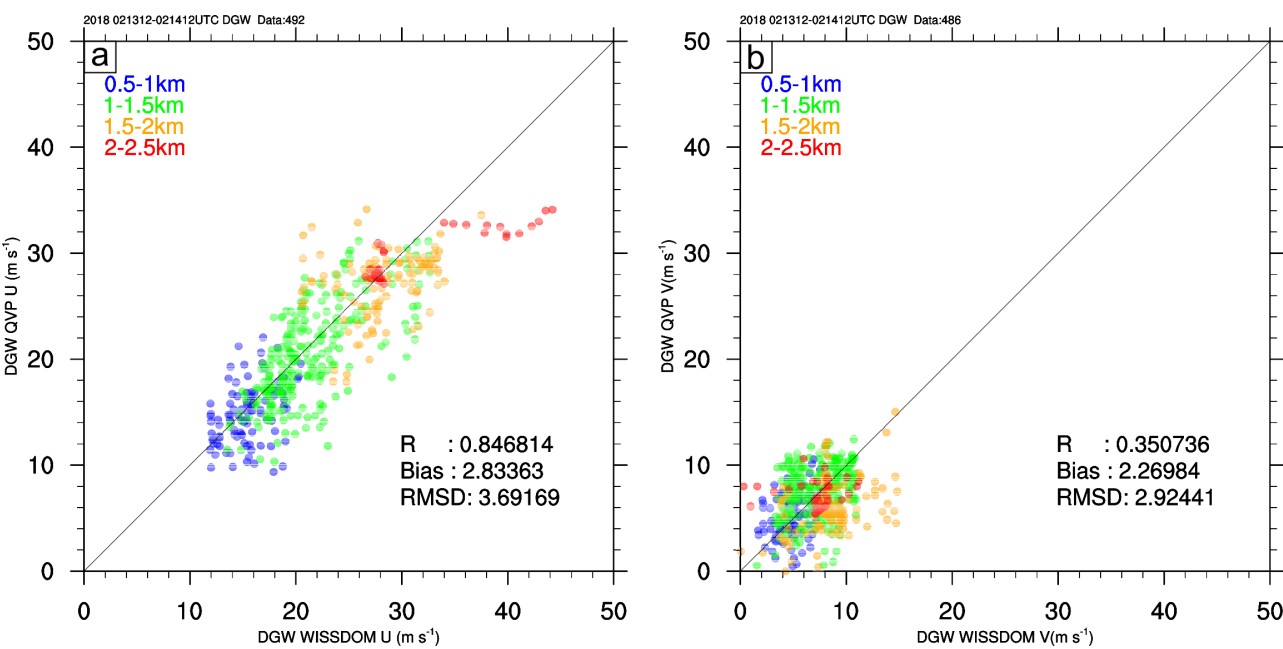

Figure 9. The same as Fig. 6 but for (a) u-component winds between the WISSDOM synthesis (x-axis) and lidar QVP (y-axis). (b) The same as (a) but for v-component winds.


Table 2 Summary of the intercomparisons between WISSDOM and observations

| | | Correlation coefficient | Average bias (m s$^{-1}$) | RMSD (m s$^{-1}$) |
|---|---|---|---|---|
| WISSDOM-sounding | u-component | 0.97 | -0.78 | 1.77 |
| | v-component | 0.87 | 0.09 | 1.65 |
| WISSDOM-lidar QVP | u-component | 0.84 | 2.83 | 3.69 |
| | v-component | 0.35 | 2.26 | 2.92 |

Compared to the sounding observations, additional w-component winds are available in
lidar QVP, which allows us to check their discrepancies in 3D winds. However, most of the
vertical velocity observations were quite weak (approximately ±0.2 m s$^{-1}$) above the DGW site,
and relatively low reliability of the derived vertical velocity could be expected in this event.
Therefore, the average vertical profiles of 3D winds were utilized to qualitatively check the
discrepancies between WISSDOM synthesis and lidar QVP during the research period (Fig. 10).
The results show that the average u-component winds have relatively smaller discrepancies
(approximately <1 m s$^{-1}$) between the WISSDOM synthesis (marked as WISS-U in Fig. 10) and
lidar QVP (marked as QVP-U) below ~1.3 km MSL at the DGW site. In contrast, there were
larger discrepancies (approximately >2 m s$^{-1}$) between 1.3 km and 2 km MSL. The average v-
component winds derived from WISSDOM (marked as WISS-V) and lidar QVP (QVP-V) were
generally weak, and the ranges of WISS-V and QVP-V were between ~2 m s$^{-1}$ and 8 m s$^{-1}$.
Generally, the vertical profiles of WISS-V were nearly overlain with QVP-V, and their
discrepancies existed in the height range 1.6~2.0 km MSL (maximum ~4 m s$^{-1}$). Smaller (larger)
discrepancies of w-component winds were significantly below (above) the height at ~1.3 km
MSL (maximum discrepancies ~0.6 m s$^{-1}$ at 1.7 km MSL). Despite the larger discrepancies, the
similar patterns of W can also be shown. In summary, the discrepancies in the 3D winds between
the WISSDOM synthesis and lidar QVP were small in the lower layers and large in the higher
layers because the observational data from lidars and AWS provided good quality and sufficient
wind information at the lower layers but not in the higher layers (lower coverages of lidar data
above 1.3 km MSL, cf. Fig. 4).

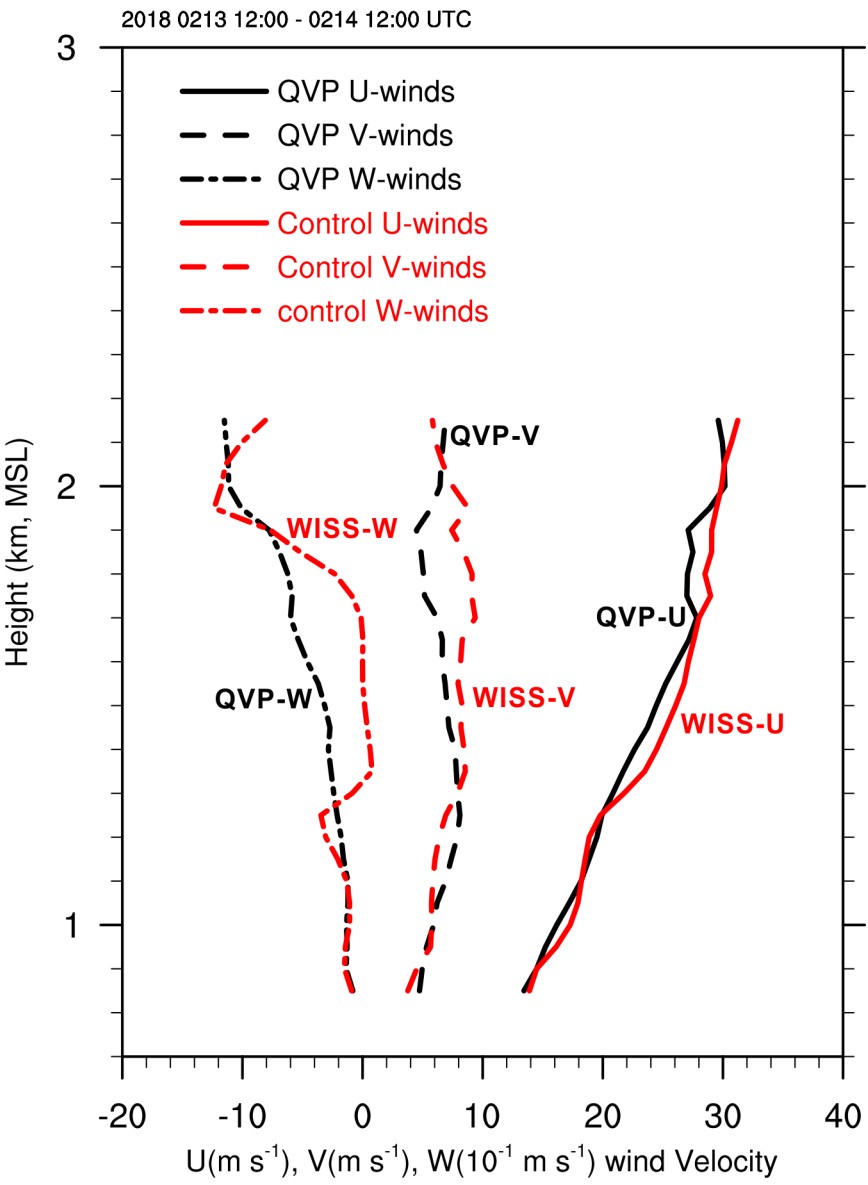

Figure 10. Vertical wind profiles of average 3D winds derived from the WISSDOM synthesis (red lines and vectors) and lidar QVP (black lines and vectors) above the DGW site from 12:00 UTC on 13 to 12:00 UTC on 14 Feb. 2018. Solid lines indicate u-component winds (m s$^{-1}$), dashed lines indicate v-component winds (m s$^{-1}$), and dash-dotted lines indicate w-component winds ($1 \times 10^{1}$ m s$^{-1}$). The u-, v-, and w-component winds derived from the WISSDOM synthesis (lidar QVP) were marked by WISS-U (QVP-U), WISS-V (QVP-V), and WISS-W (QVP-W), respectively.

Fig. 11 shows the quantile distribution of statistical errors of wind direction, wind speed and
vertical velocity between the WISSDOM synthesis and lidar QVP during the research period.
The IQR of the wind direction is smaller (−5~5 degrees) in the layers from 0.85 km to 1.5 km
MSL and turns to approximately −10~0 degrees above 1.5 km MSL. The median values of wind
direction are smaller −5~5 degrees) from near the surface to the upper layers (Fig. 11a). Fig. 11b
shows that the median values (IQR) of wind speed are approximately –1~1 m s$^{-1}$ (–2~2 m s$^{-1}$)
below 1.5 km MSL, and they all become larger with heights above 1.5 km MSL (between −1 and
3 m s$^{-1}$ for median values and −4~4 m s$^{-1}$ for the IQR). The statistical error of the vertical velocity
reveals that the IQR is −0.2~0.2 m s$^{-1}$ (−0.8~0.8 m s$^{-1}$) below (above) 1.3 km MSL, and the
median values are 0~0.2 m s$^{-1}$ −0.2~0.6 m s$^{-1}$) below (above) 1.3 km MSL. The results of
statistical errors are summarized in Table 3.

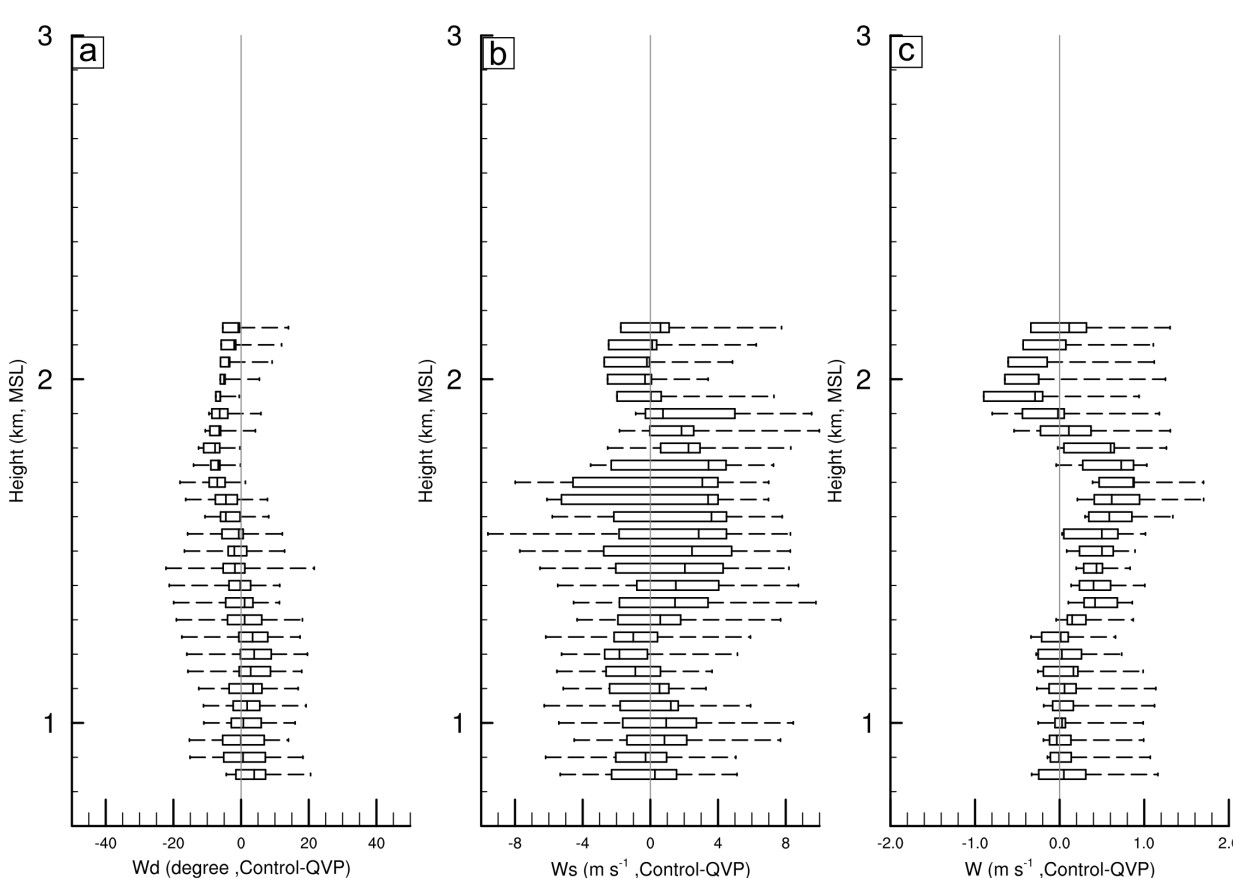


Figure 11. The box plot of average (a) wind direction discrepancies between the WISSDOM synthesis and sounding

observations above the DGW site during the research period. (b) Same as (a) but for the wind speed. (c) Same

as (a) but for the w-component winds.


Table 3     Summary of the statistical errors between WISSDOM and observations

|  |  | Interquartile range (IQR) | Median values |
|---|---|---|---|
| WISSDOM-sounding | wind direction | 0~2.5 (deg.) | 0~2.5 (deg.) |

| | | | |
|---|---|---|---|
| | wind speed | −2~0.5 (m s$^{-1}$) | −1~0.5 (m s$^{-1}$) |
| | wind direction | −10~5 (deg.) | −5~5 (deg.) |
| WISSDOM-lidar QVP | wind speed | -4~4 (m s$^{-1}$) | −1~3 (m s$^{-1}$) |
| | w-component winds | −0.8~0.8 (m s$^{-1}$) | −0.2~0.6 (m s$^{-1}$) |

## 5. Sensitivity test with various datasets, data implementation and weighting coefficients

### 5.1 Impacts of various datasets (Experiment A)

In this section, the impacts of various datasets on data implemented in the WISSDOM synthesis were evaluated. In particular, the quantitative variances between each design, control run, sounding observations, and the QVP can be estimated. The basic setting of Experiment A took off several inputs from the WISSDOM control run (cf. Table 1) as four designs in Experiment A. The details of these four designs are summarized in Table 4 as the control run without the lidar observations (A-1), the control run without the AWS observations (A-2), the control run without the sounding observations (A-3) and the control run without the LDAPS data (A-4). The discrepancies of 3D winds were examined between the control run and each design in Experiment A. Since the environmental wind speed is nearly comprised of uniform westerlies in this event, the results only show the difference in u-component winds between control run and each design (A-1~A-4) in Fig. 12. An additional test was designed, where only Doppler lidar data are used without other constraints from $J_6 \sim J_8$ (A-5) to evaluate the performances between the modified and original versions of WISSDOM.

Fig. 12a reveals the discrepancies in horizontal u-component winds at 800 m MSL as the A-1 is subtracted from the control run. This result reflects the impacts of lidar observations on the u-component winds in the WISSDOM synthesis. The most significant contributions from the

600 lidar observations are the high wind speed existing near the DGW site in a relatively narrow

601 valley. The mechanisms of the accelerated wind speed due to the channeling effect in this local

602 area were verified by our previous study (Tsai et al. 2022). The lidar observations also contributed

603 to the high wind speed in another area near the western side of the MHS site (128.68°E, 37.66°N).

604 Based on the analysis in the vertical cross section of u-component winds in A-1 (Fig. 12b), the

605 lidar observations significantly affected the high wind speed only in the lower levels (below ~900

606 m MSL) but not in the higher levels. Lidar observations provided sufficient coverage only for

607 lower levels and not higher levels (cf. Fig. 4).


Table 4 Experiment setting (sensitivity testing)

| | | |
|---|---|---|
| Control run | Various datasets | Including Doppler lidars, AWS, Soundings, LDAPS |
| | Interpolation of AWS | RI: 1.0 km, VE: 90% |
| | Weighting Coefficient | Doppler Lidars ($\alpha_1$): $10^6$<br>Background ($\alpha_2$):$10^2$<br>Sounding ($\alpha_6$): $10^6$<br>AWS ($\alpha_7$): $10^6$<br>LDAPS ($\alpha_8$): $10^3$ |
| Experiment A | Various datasets | A-1 Excluding Doppler Lidars<br>A-2 Excluding AWS<br>A-3 Excluding Soundings<br>A-4 Excluding LDAPS<br>A-5 Only Doppler lidars |
| Experiment B | Interpolation of AWS | B-1 RI: 0.5 km, VE: 50%<br>B-2 RI: 0.5 km, VE: 90%<br>B-3 RI: 1.0 km, VE: 50%<br>B-4 RI: 2.0 km, VE: 50%<br>B-5 RI: 2.0 km, VE: 90% |
| Experiment C | Weighting Coefficient (constraints) | C-1 AWS ($\alpha_7$): $10^3$<br>C-2 Doppler Lidars ($\alpha_1$): $10^3$<br>C-3 LDAPS ($\alpha_8$): $10^6$ |


610   The impacts of the AWS cause negative values on the u-component winds in most areas at

611 800 m MSL in A-2 (Fig. 12c), especially in the western areas of the MHS site. Negative

contributions of the u-component winds produced by the AWS observations were restricted near
the surface, and the low wind speed area was extended to ~100 m above the surface (Fig. 12d).
The contributions of the u-component winds from the sounding observations were weak near the
DGW sounding site in A-3 (Figs. 12e and 12f). The impacts of u-component winds from the
LDAPS datasets were rather smaller in most of analysis area. in A-4 (Figs. 12g and 12h).
Relatively weak winds were presented near the surface from the results of A-5 (Figs. 12i and
12j). These results reflect that the additional constraints play crucial roles, especially at lower
layers. Furthermore, it is implied that the winds can be reasonably retrieved when additional
constraints are set in the modified version of WISSDOM.

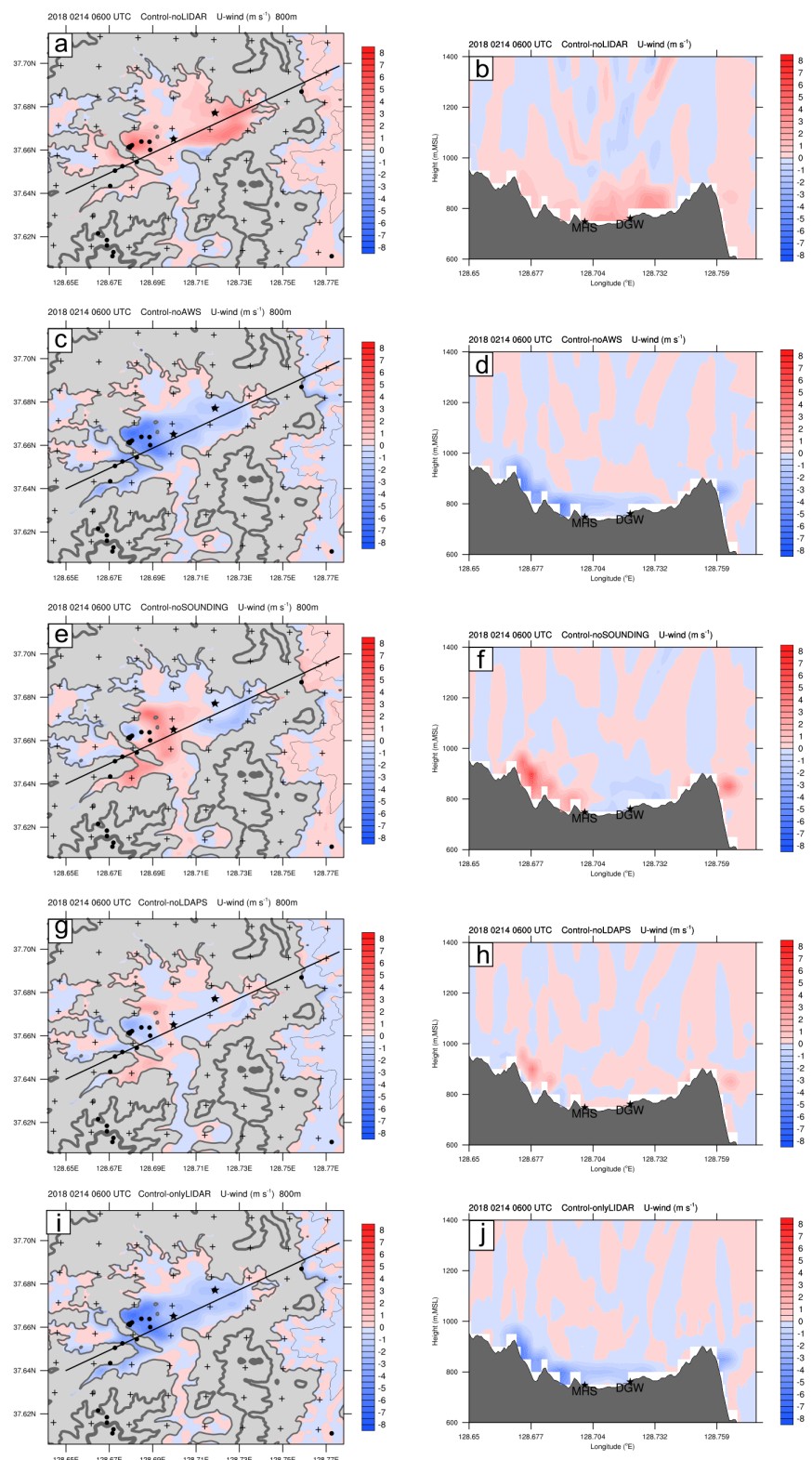


Figure 12. (a) The discrepancies in horizontal u-component winds between the control run and A-1 at 800 m MSL

at 06:00 UTC on 14 Feb. 2018. (b) The same as (a) but for the vertical section along the black line in (a). (c) and

(d) are the same as (a) and (b) but for A-2. (e) and (f) are the same as (a) and (b) but for A-3. (g) and (h) are the

same as (a) and (b) but for A-4. (i) and (j) are the same as (a) and (b) but for A-5.


Averaged discrepancies in derived 3D winds for each vertical level in entire domain are
shown in Fig. 13a. These results summarized a series of sensitivity tests if the WISSDOM
synthesis lacks certain data inputs (i.e., A-1~A-5 in Experiment A) for derived u-, v- and w-
component winds in the test domain. Overall, the maximum absolute value of averaged
discrepancies for Experiment A are smaller than approximately 0.5 m s$^{-1}$, which are the
discrepancies of the u-component winds for A-1 and A-2 located at 800 m MSL. Except for these
values, the values of the derived u-, v- and w-component winds for A-1~A-2 are approximately
smaller than 0.2 m s$^{-1}$ from the surface up to the top in the test domain. Based on the results of
A-5, relatively stronger values of derived u-component (exceeded $-0.4$ m s$^{-1}$ at lower layers) can
be obtained from the setting like old version of WISSDOM. The wind speed can be better
modulated in modified version of WISSDOM when the Doppler lidar observations were adopted.
In addition, the discrepancies in derived 3D winds between sounding observations and QVP
were also examined along the sounding tracks (Fig. 13b) and above the DGW site (Fig. 13c).
Sounding observations played an essential role in the derived winds along its tracks. The
maximum discrepancies of u-component winds are exceeded by approximately $-2$ m s$^{-1}$ , and v-
component winds are exceeded by approximately $-1$ m s$^{-1}$ if the WISSDOM synthesis lacks
sounding observations. However, small discrepancies (nearly 0 m s$^{-1}$) were presented when the
sounding data were implemented, and the lidar was not implemented at all levels in A-1. The
peaks in the discrepancies manifested the potential impacts from the lidar and AWS. This may
result from lidar and AWS having higher data coverage at ~1.4 and 0.8 km MSL, respectively
(cf. Fig. 4). The discrepancies of sounding observation and control run in u- and v-component
winds reveal relatively small values than the A-3 but similar to the other designs (purple lines in
Figs. 13b). The maximum discrepancies between the derived winds and the QVP winds are
approximately $-4$ and 4 m s$^{-1}$ associated with u- and v-component winds, and $-1$ and 0 to the w-
component winds. Generally, the results reveal similar trends in A-1~A-5, implying that all the
inputs in the WISSDOM synthesis are equally significant against the QVP. The QVP winds and
control run discrepancies in u- and v-component winds show similar values for all designs, but
relatively small values can be obtained in w-component winds (purple lines in Figs. 13c). In
summary, the results of this experiment (cf. Fig. 13) show that the lidar, sounding, and AWS data
are more critical inputs than the LDAPS in modified WISSDOM. Therefore, it will be beneficial
if various inputs can be included in the synthesis.

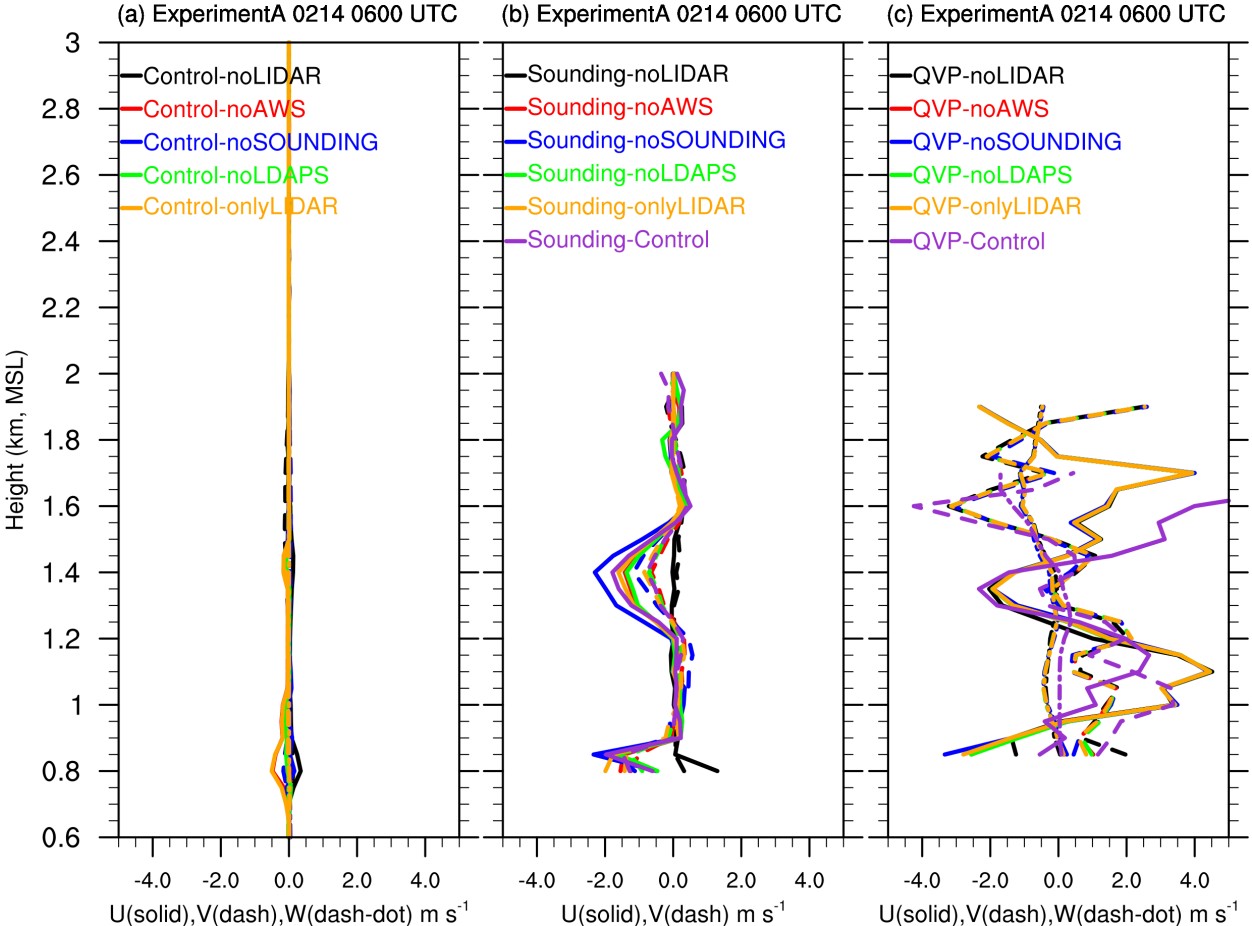

Figure 13. (a) Vertical profiles of averaged discrepancies of 3D winds for each design in Experiment A at 06:00

UTC on 14 Feb. 2018. The averaged discrepancies of u-, v- and w-component winds were plotted by solid, dash,

and dash-dot lines, and the black, red, blue, green and orange lines indicate A-1, A-2, A-3, A-4 and A-5,

respectively. (b) The same as (a) but for the discrepancies of sounding observations and u-, and v-component

winds and control run (purple lines). (c) The same as (b) but for the discrepancies of QVP and w-component

winds.

**5.2 Radius of influence (RI) and vertical extension for the AWS (Experiment B)**

Experiment B was performed to check the discrepancies in 3D winds between the control run

and the different settings of RI and VE with the AWS observations. Because the average distance
is approximately 0.1 to 2 km between each AWS site, there were five designs (B-1~B-5) in
Experiment B with ranges of RI (VE) between 0.5 km (50%) and 2 km (90%). The details are
shown in Table 4. The horizontal u-component winds at 800 m MSL and the vertical structure of
Experiment B at one time step (06:00 UTC on 14 February 2018) are shown in Fig. 14. An
unusual circular area with positive discrepancies around the MHS site was depicted in B-1 (Figs,
14a and 14b), which may have been produced by the insufficient RI distance and VE (the circular
artefact is removed when increasing VE to 90%). Relatively smaller RI and VE values can only
include relatively less wind information if the distances are large between each AWS. Enlarging
the RI and VE are required to appropriately include more wind information from the AWS
observations. Figs. 14c and 14d show the results of B-2 as VE reached 90%. Although the unusual
circle vanished, there were discontinuities with negative values near the northern and southern
areas of the MHS site and positive areas surrounding the AWS (128.68°E, 37.66°N). The setting
of B-3 was similar to that of the control run except that the VE was 50%. The discrepancies were
relatively small, albeit dense AWS contributed even smaller negative values in the western areas
of the MHS sites (Figs. 14g and 14h). Obviously, positive discrepancies appeared near the
northern and southern areas of the MHS site in B-4 and B-5 (Figs. 14g-j). The impacts of the
AWS with various settings (B-1~B-5) on the discrepancies in u-component winds were both
restricted near the surface, even with a larger RI and high VE.

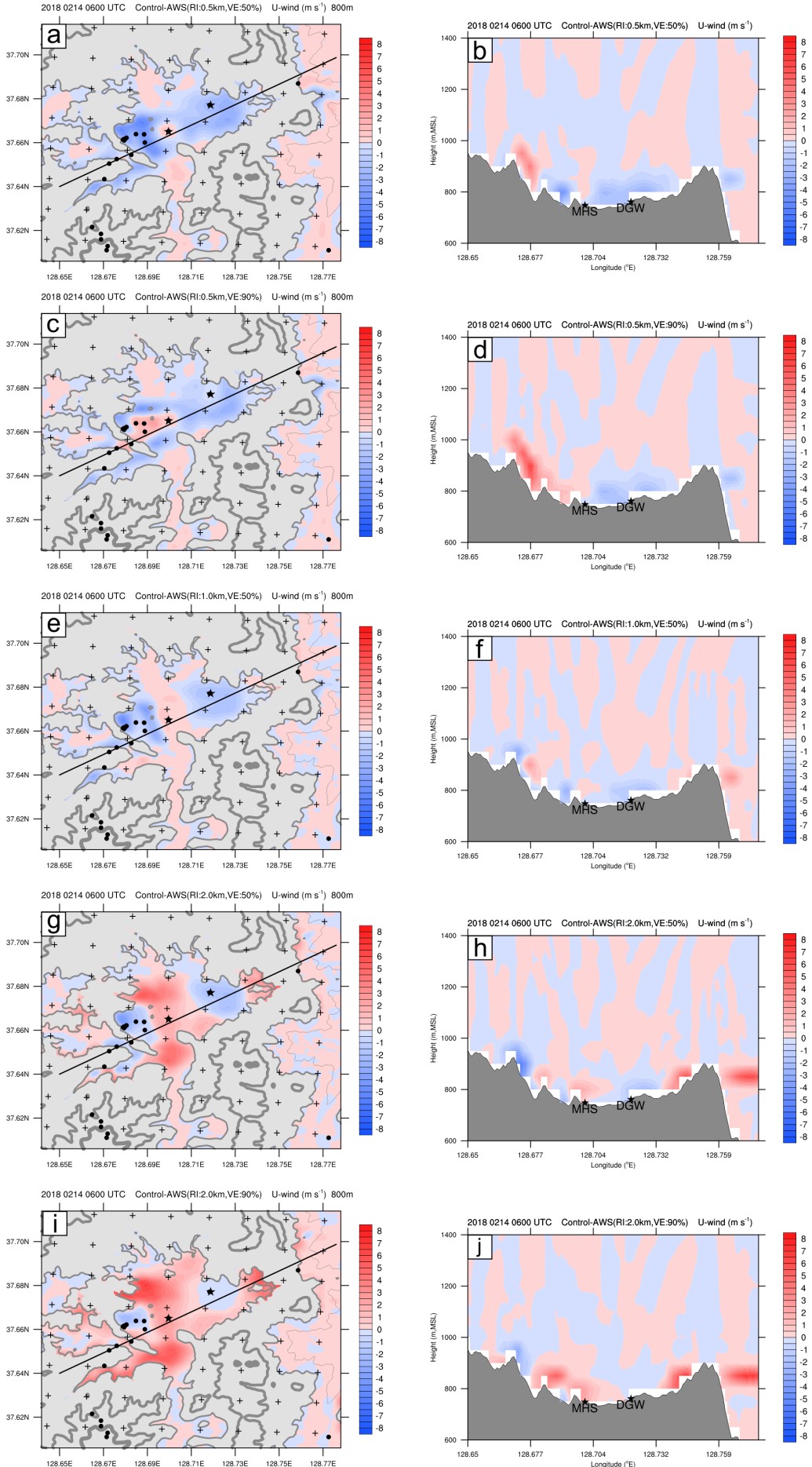


Figure 14. The same as Fig.12, but (a) and (b) for B-1. (c) and (d) are the same as (a) and (b) but for B-2. (e) and (f)
are the same as (a) and (b) but for B-3. (g) and (h) are the same as (a) and (b) but for B-4. (i) and (j) are the same
as (a) and (b) but for B-5.
Fig. 15a shows the vertical profiles of averaged discrepancies of derived 3D winds in
Experiment B. This figure summarizes the results of sensitivity testing with different settings of
the RI and VE in WISSDOM (i.e., B-1~B-5 in Experiment B, shown in Table 4) for derived u-,
v- and w-component winds in the test domain. The maximum discrepancies of u-component
winds in B-1, B-2 and B-3 were quite small at only 0.4, 0.3 and 0.2 m s$^{-1}$, respectively.
Nevertheless, the maximum discrepancies of u-component winds for B-4 and B-5 were larger
than 0.6 m s$^{-1}$ and even exceeded ~1 m s$^{-1}$. Although the discrepancies in the u-component winds
in B-1 were small, the discrepancies in the v-component winds in B-1 reveal unusual patterns,
with larger positive values at ~1100 m MSL and negative values at ~1800 m MSL (black dashed
line in Fig. 15a), the possible reason is the minimizations of cost function are not converged well
because relatively few and weak v-component winds were included in B-1. Except for this value,
the maximum discrepancies of v-component winds were small for B-2~B-5, and the maximum
discrepancies of w-component winds were also small for all of Experiment B. Note that B-3
always has the smallest discrepancies with the derived 3D winds because the setting is quite
similar to the control run. Figs. 15b and 15c show the discrepancies of derived 3D winds between
the sounding observations, QVP, and control run. Their patterns are similar to A-1~A5 (cf. Figs.
13b and 13c), except there were relatively larger values of u- (v-) component winds at lower
layers (approximately −3 and 1 m s$^{-1}$) in B-1 (Fig. 15b). The v-component winds also presented
larger values (exceeded ~3 m s$^{-1}$) below ~1.2 km MSL compared with the QVP (Fig. 15c). The
conclusions indicate that the moderate setting (i.e., RI is 1 km) would be helpful to obtain smaller
differences with the control run, sounding observations, and the QVP in this case. On the other
hand, the limited setting in experiment B (i.e., B-1) was not suitable. In addition, the wind
directions and speed should be dominated by terrain, and the implementation of AWS data is
crucial for the modified WISSDOM synthesis, especially in the lower layers.

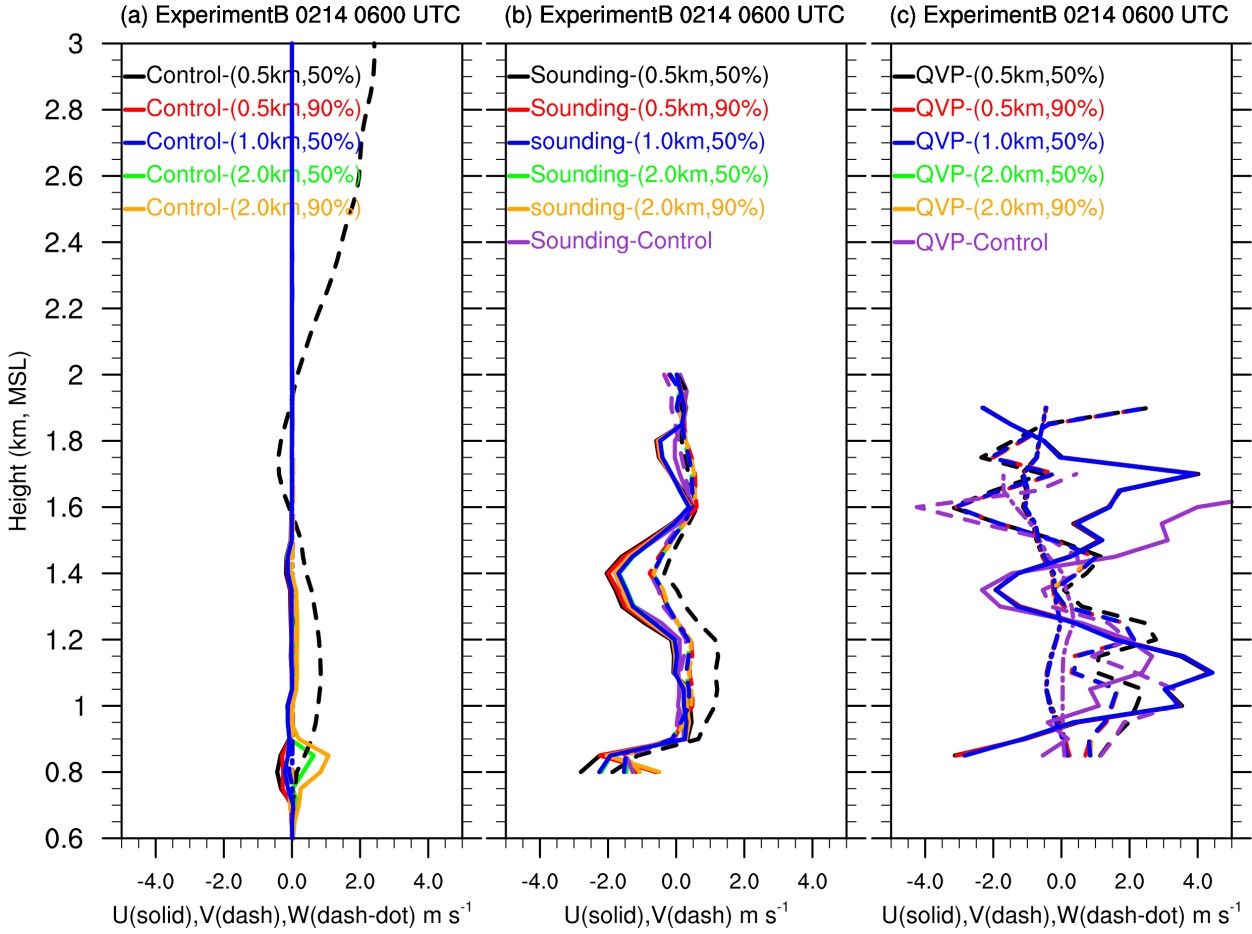


Figure 15. The same as Fig. 13. but for B-1~B-5.

## 5.3 Different weighting coefficients for the constraints (Experiment C)

Experiment C was designed to check the discrepancies in the derived u-component winds
between the control run and experimental runs with different weighting coefficients for each
constraint related to the AWS, lidar and LDAPS (corresponding to C-1, C-2 and C-3 in Table 4).
Originally, the weighting coefficients for the AWS and lidar observations were set to $10^6$, and the
value was $10^3$ for the LDAPS dataset (i.e., control run, Table 1). The results of Experiment C
show significant negative discrepancies in u-component winds near the surface in C-1, especially
in the areas next to the AWS (128.68°E, 37.66°N). The discrepancies for C-1 (Figs. 16a and 16b)
and C-2 (Figs. 16c and 16d) are similar to those for A-2 (Figs. 12c and 12d) and A-1 (Figs. 12a
and 12b), respectively. The inputs of AWS and lidar both contributed relatively weak impacts to
the WISSDOM synthesis when the weighting coefficient was set to $10^3$. Irrational patterns were
depicted when the weighting coefficient of LDAPS inputs increased to $10^6$, and larger and
positive discrepancies were crowded into most areas in the valley (i.e., C-3, Figs. 16e). Larger
and positive discrepancies existed only near the surface, and there were negative discrepancies
between approximately 1000 m and 1400 m (Fig. 16f). Significant differences often exist in
between the observations and reanalysis dataset due to the differing spatio-temporal resolutions.
The results of scenario C-3 do not converge well because there was a relatively more significant
gradient between each input as their weighting coefficients were set to be the same (i.e., $10^6$). In
this way, the effects of poor convergences might be amplified with the AWS and lidar
observations along the sounding tracks. This may be a possible reason that artificial signals
existed over the DGW site in scenario C-3.
The vertical profiles of averaged discrepancies of derived 3D winds in Experiment C are
shown in Fig. 17a. Absolute values of the discrepancies in the u-, v- and w-component winds are
smaller than 1 m s$^{-1}$, except for the discrepancies in the v-component winds with low weighting
of the AWS observations (i.e., C-1) and the discrepancies in the u- and v-component winds with
the high weighted LDAPS (i.e., C-3). The discrepancies in the v-component winds in C-1
exceeded $-5$ m s$^{-1}$ at ~1100 m MSL and were larger than $-15$ m s$^{-1}$ above 2600 m MSL. These
unreasonable characteristics are also shown as the discrepancies in the v-component winds in B-
1 (cf. Fig. 15a). The discrepancies in the u- and v-component winds in C-3 are 15 m s$^{-1}$ and 4 m
s$^{-1}$, respectively, in the layers between 700 and 900 m MSL. Alternative positive and negative
discrepancies in the range of $-3$ to 3 m s$^{-1}$ for the u-component winds in C-3 were found above
1000 m MSL.
The discrepancies in between the derived 3D winds in Experiment C and the sounding
observations and QVP, respectively, were also examined. Compared to the sounding
observations, more significant discrepancies in the u- and v-component winds (exceeding ~20 m
s$^{-1}$) can be obtained when reducing the weighting coefficients of the AWS and increasing the
weighting coefficients of the LDAPS data (Fig. 17b). However, the impacts of lidar against the
QVP are shown; their discrepancies are in the range of −1 to 2 m s$^{-1}$ for the u-component winds
in C-2 (Fig. 17c). The discrepancies of sounding observations, the QVP winds, and the control
run were more minor than all designs in Experiment C (purple lines in Figs. 17b and 17c). The
conclusions reveal that the weighting coefficients of the AWS and LDAPS are significantly
sensitive to the derived winds, and the lidar is moderately sensitive to the retrieved winds.
Therefore, the weighting coefficients of LDAPS and AWS are better to be $10^3$ and $10^6$ in this
case.

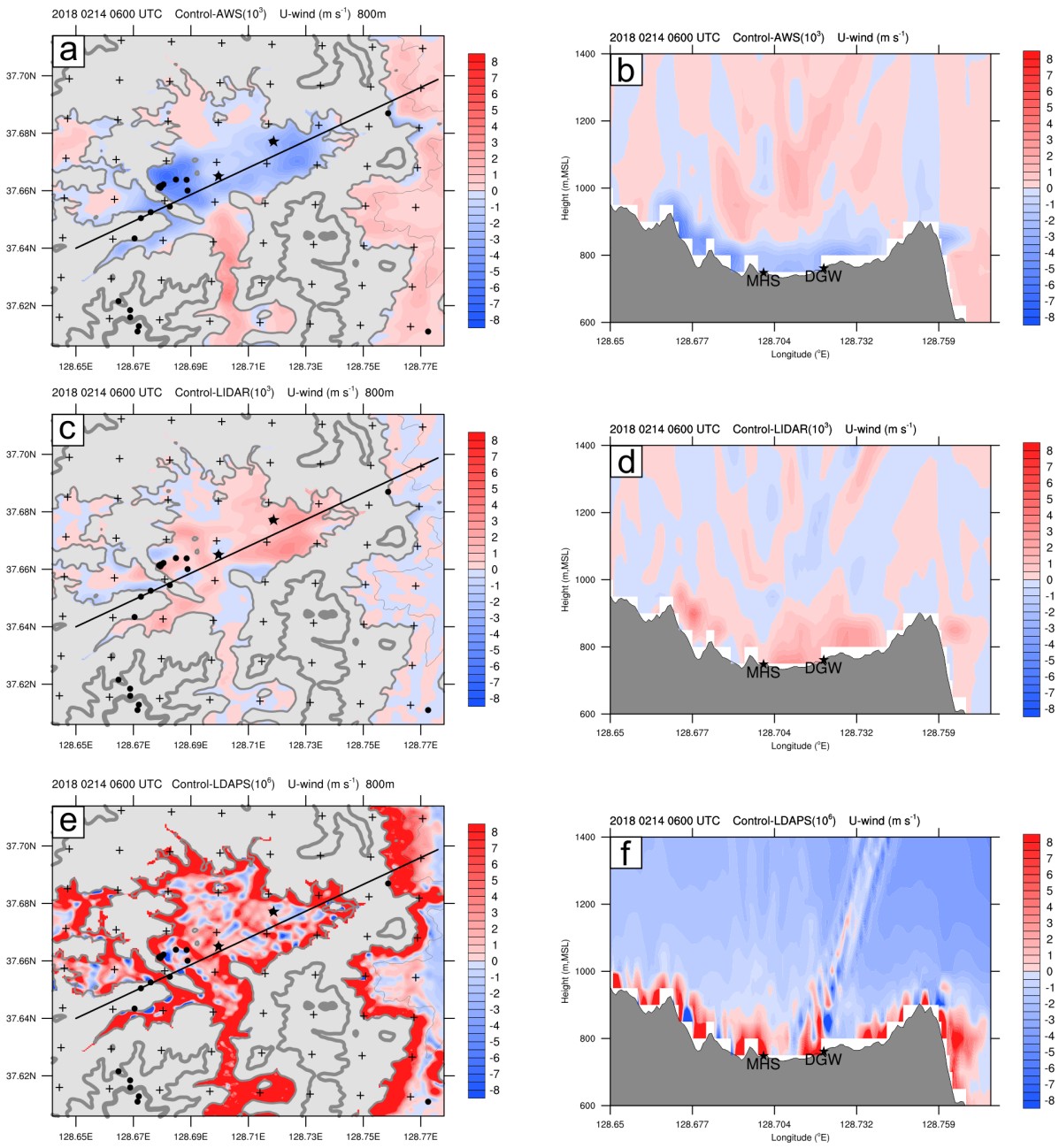

Figure 16. The same as Fig.12, but (a) and (b) for C-1. (c) and (d) are the same as (a) and (b) but for C-2. (e) and (f) are the same as (a) and (b) but for C-3.

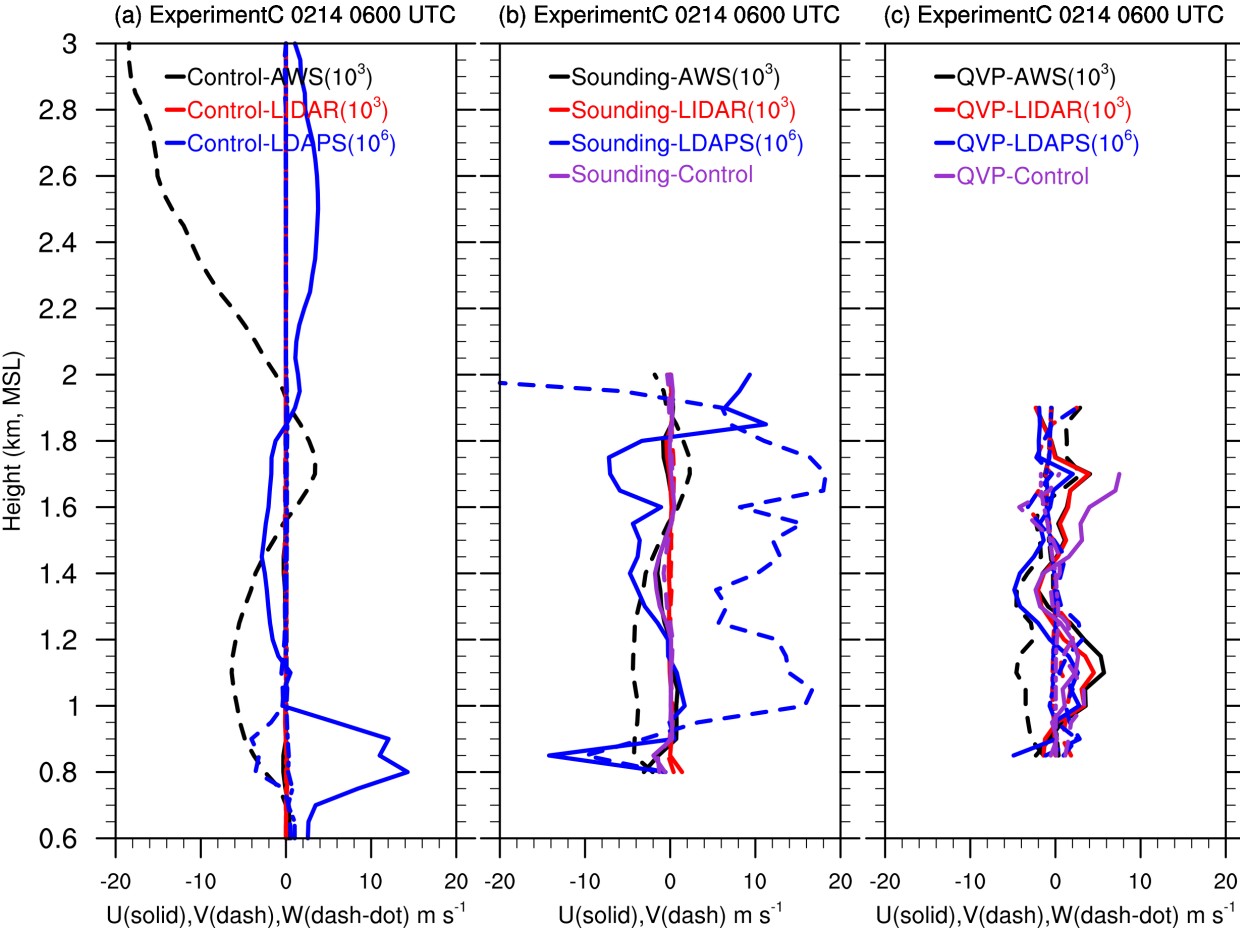

Figure 17. The same as Fig. 13 but for C-1~C-3.

## 6. Conclusion

A modified WISSDOM synthesis scheme was developed to derive high-quality 3D winds under clear-air conditions. The main difference from the original version is that multiple lidar observations were used in the modified version, replacing radar data. High-resolution 3D winds (50 m horizontally and vertically) were first derived in the modified WISSDOM scheme. In addition, the wind information was separated from the background in the modified version. Therefore, all available datasets were included as one of the constraints in the cost function in this study. The data implementation and the detailed principles of the modified WISSDOM were

also elaborated. This modified WISSDOM scheme was performed over the TMR to retrieve 3D
winds during a strong wind event during ICE-POP 2018. The performance was evaluated via a
series of sensitivity tests and compared with conventional observations.
The intercomparisons of horizontal winds during the entire research period reveal a relatively
high correlation coefficient between the optimal results of WISSDOM synthesis and sounding's
u- (v-) component winds exceeding 0.97 (0.87) at the DGW site. Furthermore, the average bias
is $-0.78$ m s$^{-1}$ (0.09 m s$^{-1}$), and the RMSD is 1.77 m s$^{-1}$ (1.65 m s$^{-1}$) for the u- (v-) component
winds. The intercomparisons of 3D winds between the WISSDOM synthesis and lidar QVP also
showed a higher correlation coefficient (0.84) for u-component winds, but a relatively smaller
correlation coefficient remained at 0.35 for v-component winds in this strong wind event. The
average bias (RMSD) of u-component winds is 2.83 m s$^{-1}$ (3.69 m s$^{-1}$), and the average bias and
RMSD of v-component winds are 2.26 m s$^{-1}$ and 2.92 m s$^{-1}$, respectively (cf. Table 2). Chen
(2019) analyzed the correlations between 3D winds derived from radar and observations in
several typhoon cases; the mean correlation coefficient ranged from 0.56 to 0.86, and the RMSD
was between 1.13 and 1.74 m s$^{-1}$. Compared to their results, only u-component winds have
relatively higher correlation coefficients, but the RMSD values are slightly higher in this study,
which may have been caused by the high variability in westerly winds associated with the moving
LPS. The statistical error results of the winds between the optimal results of WISSDOM synthesis
and observations show a good performance of the retrieved 3D winds in this strong wind event
(Table 3). Generally, the median values of wind directions are within ~10 degrees. Compared
with lidar QVP (sounding observations) above the DGW site, the median values of the wind
speed are approximately $-1$~3 m s$^{-1}$ ($-1$~0.5 m s$^{-1}$), and the vertical velocity is within $-0.2$~0.6
m s$^{-1}$; the IQR of wind directions is $-10$~5 (0-2.5) degrees, the wind speed is approximately $-4$~4
m s$^{-1}$ ($-1$~3 m s$^{-1}$), and the vertical velocity is $-0.8$~0.8 m s$^{-1}$. The summaries of the correlation
coefficients, average bias, the RMSD, and range of statistical errors are shown in the schematic
diagrams as Figs. 18a and 18b.

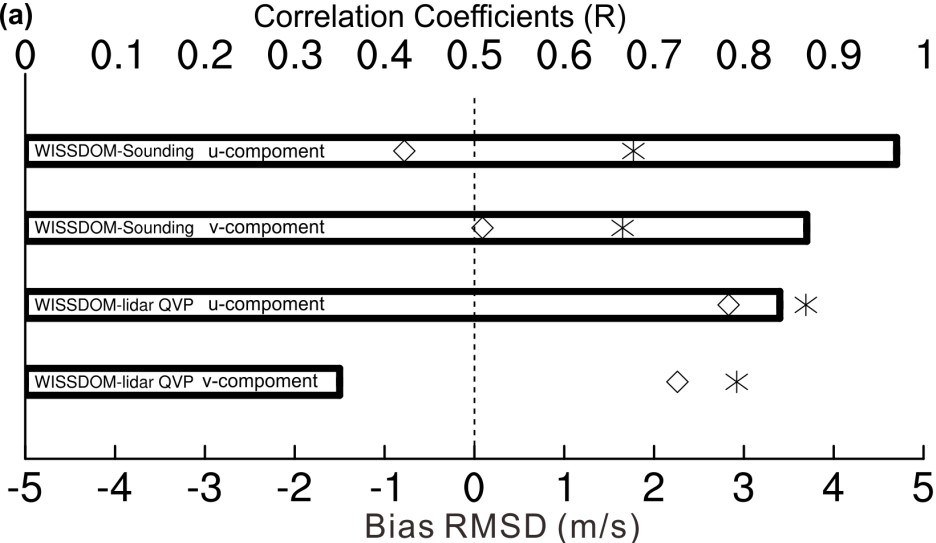

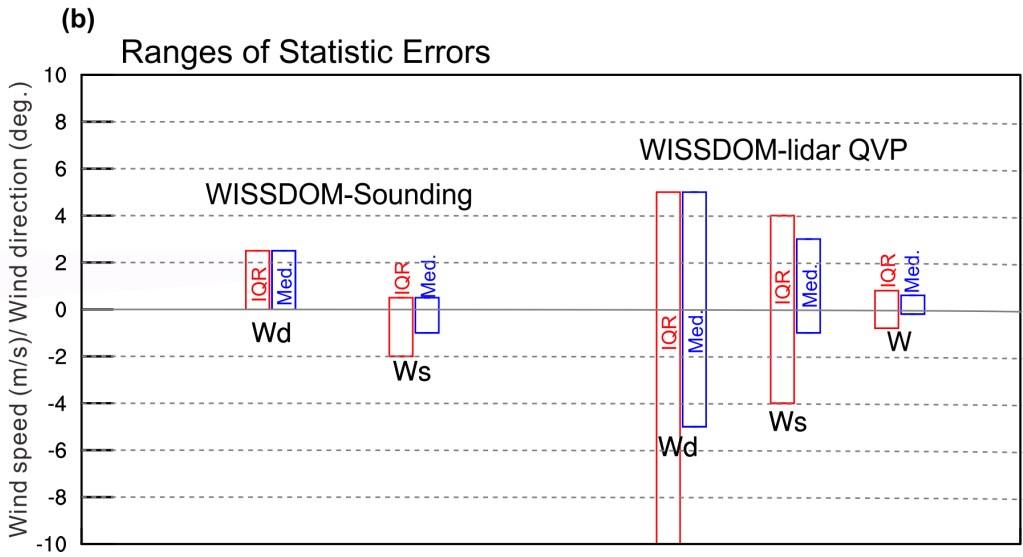

Figure 18. Schematic diagrams for the results of intercomparisons on (a) the correlation coefficients (R, histograms),
the average bias (marked as diamonds), and the RMSD (marked as asterisks). (b) The ranges of statistic error for
the IQR (red boxes) and median values (blue boxes). The wind directions, wind speed and w-component winds
are denoted as Wd, Ws, and W respectively.
A control run (see the basic setting in Table 1) was set to explore the importance of acquired
observation datasets, various distances of RI, VE from the AWS observations, and the weighting
coefficient for each constraint (i.e., Experiments A-C, Table 4). The results of Experiment A
show that the lidar and AWS play critical roles in the derived horizontal winds, and the lidars
(AWS) provided positive (negative) contributions in stronger (weaker) wind speeds near the
surface. The sounding and the LDAPS provided relatively smaller impacts on the derived
horizontal winds from the WISSDOM synthesis. In Experiment B, minor discrepancies in 3D
winds were depicted when the RI (VE) was set to 1 km (50%), which indicated that the optimal
setting of the RI is 1 km. However, there were larger discrepancies in 3D winds (from $-0.4$ m s$^{-1}$
to ~1 m s$^{-1}$) when the RI was set at 0.5 km and 2 km, and the VE was set between 50% and 90%
(cf. Fig. 15). In Experiment C, significant discrepancies in 3D winds appeared by decreasing
(rising) the weighting coefficient from the AWS observations (LDPAS datasets). Relatively
reasonable winds can be derived with the setting of 1 km in RI and 90% in VE over complex
terrain (i.e., the same setting as the control run). These sensitivity tests will help verify the impacts
against various scenarios and observational references in this area.

This study demonstrated that reasonable patterns of 3D winds were derived by the modified

WISSDOM synthesis scheme in a strong wind event. Reasonable winds can be retrieved from
modified WISSDOM with sufficient coverage from the data, a moderate weighting function, and
appropriate implementation from different datasets. In the future, many cases are required to
check the performance of this modified WISSDOM scheme with different synoptic weather
systems under clear-air conditions in different seasons. In addition, knowing the detailed
kinematic fields will help us to identify where the flow accelerates/decelerates over complex
terrain. Thus, the possible mechanisms of extremely strong winds in South Korea will be well
documented through combinations with derived dynamic fields (Tsai et al., 2018, 2022),
thermodynamic fields (Liou et al., 2019), observations and simulations. The detailed wind
structures can be well documented for any meteorological phenomena in clear-air conditions
(e.g., land−sea breezes, microdownbursts, nonprecipitation low-pressure systems, etc.) via a
modified version of WISSDOM. It also has broad applications in site surveys of wind turbines,
wind energy, monitoring wildfires, outdoor sports in mountain ranges, and aviation security.
*Code and data availability.* The scanning Doppler lidars, AWS, and sounding data used in this
study are available through zenodo: https://doi.org/10.5281/zenodo.6537507. The LDAPS
dataset is freely available from the KMA website (https://data.kma.go.kr).
*Acknowledgments*. This work was supported by the National Research Foundation of Korea (NRF)
grant funded by the Korea government (MSIT) (No. 2021R1A4A1032646) and by the Korea
Meteorological Administration Research and Development Program under Grant KMI2022-
840 00310.

*Author contributions.* This work was made possible by contribution from all authors.
Conceptualization, CLT, GWL; methodology, CLT, YCL, and KK; software, CLT, YCL, and
KK; validation, KK, YCL, and GWL; formal analysis, CLT, and KK; investigation, CLT, and
GWL; writing—original draft preparation, CLT; writing—review and editing, GWL, YCL and
KK; visualization, CLT; supervision, GWL; funding acquisition, GWL. All authors have read
and agreed to the published version of the manuscript.
*Competing interests.* The authors declare that they have no conflict of interest.
*Special issue statement.* This article is part of the special issue "Winter weather research in
complex terrain during ICE-POP 2018 (International Collaborative Experiments for
Pyeongchang 2018 Olympic and Paralympic winter games) (ACP/AMT/GMD inter-journal SI)".
It is not associated with a conference.

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
