# Peer review of "High Resolution 3D Winds Derived from a Modified WISSDOM Synthesis"

_Atmospheric Measurement Techniques, 2022_

## Referee Comment (RC1)

[referee-annotated manuscript omitted]

---

## Author Response (AR1)

**JGR-2022-218**
**Responses (highlighted with red) to Referee#1**
**14 Oct. 2022**

Review "High Resolution 3D Winds Derived from a Newly Developed WISSDOM Synthesis Scheme using Multiple Doppler Lidars and Observations"

**Summary:**

The study introduces a new version of the WISSDOM algorithm. In comparison to the previous version, multiple wind datasets can be incorporated as independent input sources. In addition, here, Doppler Lidar data is used instead of Doppler radar data. This allows the algorithm to operate in clear-sky conditions. With a case study, the WISSDOM results are compared to wind measurements and sensitivity tests are performed on key features of the algorithm.

**General remarks:**

The overall goal of the study could be presented more clearly. The authors present a new version of the WISSDOM algorithm; however do not compare it to the performance of the old version. There is also very little discussion about the change from Doppler radar to Doppler LIDAR data (issues, benefits, etc.). Some aspects of the algorithm could be explained more clearly – the goal should be that the algorithm can be reproduced independently with the information given.

We appreciate Referee#1 providing helpful and insightful comments, which help us to improve the manuscript substantially. A set of responses to your comments is provided below. Specific locations of modified portions (marked as underlines) were also noted as the number of lines in the revised manuscript.

We have checked your comments carefully and emphasized the main goal of this study in Introduction section; more precise words for the new version (modified) of the WISSDOM were also defined. In addition, we have clarified the benefits and the reasons why we implanted the Doppler lidar data, but not uses Doppler radar data in our new version of WISSDOM. However, we cannot directly evaluate the performance of this new version compared with the old version under clear-air conditions, because the original design is used radar observations only in the old version. Instead of the radar observations were adopted in the old version, we performed a new design in Experiment A (A-5) that is only lidar observations are used without additional constraints (i.e., $J_6$, $J_7$, and $J_8$). Based on the idea as the retrieved winds of control run are the optimal results since it is an analytic expression in WISSDOM. Therefore, the results of the A-5 can provide a reference on the performance and discrepancies between the new and old versions of WISSDOM. Finally, we have improved the descriptions and kept the most contents in Methodology section, then the algorithm should be reproduced independently with the given information in our revisions. Please check the details as follows.

1. Because the new version of WISSDOM had considered various inputs, flexible constraints, and higher spatial resolution based on the original version of

WISSDOM. To more fit the definitions on these purposes, we would like to suitably alter the words from "newly developed" to "modified" WISSDOM synthesis scheme in the title and throughout the manuscript.

2.  The main goal of this study is to use Doppler lidar observations to retrieve high-resolution 3D winds over terrain under clear-air conditions via WISSDOM. Prior our works have been done, the Doppler lidar observations never applied in WISSDOM synthesis because the Doppler radar data is one of default inputs in the original version of WISSDOM. However, the detailed wind features are also quite important to the initiations of precipitation systems before rain or snow formatted, the reliability and performance of retrieved 3D winds should necessarily be evaluated under clear-air conditions. More clear descriptions and discussions about the context of our main goal and the benefits of Doppler lidar observations have been added in the last two paragraphs in the Introduction section and last paragraph in Section 2.2 as:

    L103-109. However, the original WISSDOM only provided 3D winds under precipitation conditions, and it cannot work well under clear-air conditions because the Doppler radar is not easy to detect radial velocity without precipitation particles. To obtain high-quality 3D winds under clear-air conditions for investigating the initiations of precipitation systems in advance of rain and snow formatted. The radial velocity observed from the scanning Doppler lidars can be used in WISSDOM, which is the most important benefit rather than Doppler radar in related research topics.

    L119-121. It is because the Doppler lidar had high spatial resolution in between 40 and 60 m horizontally, however, the Doppler radar had relatively low spatial resolution from approximately 100 to 1000 m.

    L123-125. In summary, the main goal of this study is to use Doppler lidar observations to retrieve high-resolution 3D winds over terrain under clear-air conditions via WISSDOM.

    L233-234. Although there were quite a few studies by using Doppler radar in WISSDOM, this study is first time to apply the Doppler lidar data in WISSDOM.

3.  We have emphasized that the retrieved winds from the control run are the optimal results (i.e., the analytic expression of variational-based scheme in WISSDOM) in the texts as below:

    L133-135. The 3D winds were derived by variationally adjusted solutions to satisfy the constraints in the cost function, thus the results of retrieved winds were the analytic expression in this scheme.

    L395-397. Therefore, the retrieved winds from the control run can be treated as the optimal results (i.e., analytic expression of variational-based scheme) in WISSDOM.

4.  The modified version of WISSDOM had relatively good performance, and it can be verified by performing a new design in Experiment A (A-5). The A-5 is just only

included Doppler lidar data in WISSDOM without the constraints from $J_6 \sim J_8$, this design is quite like the setting of original version of WISSDOM. The setting of A-5, test results, and the detailed descriptions about these evaluations have been added in the table 4, modified Figs. 12 and 13 in the texts as below:

L568-571. In addition, to evaluate the performances between the modified WISSDOM and original version by using Doppler lidar data, additional test was designed as only Doppler lidar data are used without additional constraints from $J_6 \sim J_8$ (A-5).

Table 4 Experiment setting (sensitivity testing)

| | | |
|---|---|---|
| Control run | Various datasets | Including Doppler lidars, AWSs, Soundings, LDAPS |
| | Interpolation of AWS | RI: 1.0 km, VE: 90% |
| | Weighting Coefficient | Doppler Lidars ($\alpha_1$): $10^6$
Background ($\alpha_2$):$10^2$
Sounding ($\alpha_6$): $10^6$
AWS ($\alpha_7$): $10^6$
LDAPS ($\alpha_8$): $10^3$ |
| Experiment A | Various datasets | A-1 Excluding Doppler Lidars
A-2 Excluding AWSs
A-3 Excluding Soundings
A-4 Excluding LDAPS
A-5 Only Doppler lidars |
| Experiment B | Interpolation of AWS | B-1 RI: 0.5 km, VE: 50%
B-2 RI: 0.5 km, VE: 90%
B-3 RI: 1.0 km, VE: 50%
B-4 RI: 2.0 km, VE: 50%
B-5 RI: 2.0 km, VE: 90% |
| Experiment C | Weighting Coefficient (constraints) | C-1 AWS ($\alpha_7$): $10^3$
C-2 Doppler Lidars ($\alpha_1$): $10^3$
C-3 LDAPS ($\alpha_8$): $10^6$ |

L592-595. Relatively weak winds were presented from the results of A-5 (Figs. 12i and 12j), especially at the lower layers. These results reflects that relatively stronger winds were retrieved when additional constraints are removed. Furthermore, it is also implied that the retrieved winds can be reasonably adjusted in the modified version of WISSDOM.

[Figure]

Figure 12. (a) The discrepancies in horizontal u-component winds between the control run and A-1 at 800 m MSL at 06:00 UTC on 14 Feb. 2018. (b) The same as (a) but for the vertical section along the black line in (a). (c) and (d) are the same as (a) and (b) but for A-2. (e) and (f) are the same as (a) and (b) but for A-3. (g) and (h) are the same as (a) and (b) but for A-4. (i) and (j) are the same as (a) and (b) but for A-5.

[Figure]

L609-612. Based on the results of A-5, relatively stronger values of derived u-component (exceeded −0.4 m s$^{-1}$ at lower layers) can be obtained from the setting like old version of WISSDOM. The wind speed can be better modulated in modified version of WISSDOM when the Doppler lidar observations were adopted.

Figure 13. Vertical profiles of averaged discrepancies of 3D winds for each design in Experiment A at 06:00 UTC on 14 Feb. 2018. The averaged discrepancies of u-, v- and w-component winds were plotted by solid, dash, and dash-dot lines, and the black, red, blue, green, and orange lines indicate A-1, A-2, A-3, A-4 and A-5, respectively.

The validation of the control run is performed with sounding data and LIDAR quasi-vertical profiles. As both sounding data and LIDAR data are used in WISSDOM, this is not an independent verification and rather reflects on the importance of these data sources in combination with the others.

Thank you for this comment. In our study, we try to evaluate more about the performance of the optimal results from WISSDOM synthesis, so we performed the intercomparisons between control run and observations. The results indicates that the RMSD of u- and v-component winds between control run and sounding observations are acceptable (1.65-1.77 m s$^{-1}$) since the values are quite close to the evaluations in previous study (1.13-1.74 m s$^{-1}$, Chen, 2019).
Furthermore, we designed a series sensitivity testing to understand how many impacts from each independent observation. For example, there were relatively good performance of derived u-, and v-component winds compared with sounding observations (cf. Fig. 6a), and very small impacts (~0.1 m s$^{-1}$) can be seen if the sounding data was not included (cf. Fig. 13). These results implied that the ranges of errors are relatively small when we can evaluate the discrepancies between control run and independent sounding observations. The descriptions about these statements have been added and modified for clear as below:

L438-439. Detailed analyses were performed in this section to quantitatively evaluate the accuracy of the optimally derived 3D winds from the WISSDOM synthesis.

L612-614. These results also implied that the ranges of errors are relatively small when we try to evaluate the discrepancies between the control run and each independent observation.

L 712-714. The intercomparisons of horizontal winds during the entire research period reveal a relatively high correlation coefficient between the optimal results of WISSDOM synthesis and sounding's u- (v-) component winds exceeding 0.97 (0.87) at the DGW site.

L726-728. The statistical error results of the winds between the optimal results of WISSDOM synthesis and observations show a good performance of the retrieved 3D winds in this strong wind event (Table 3).

The sensitivity tests do not present a clear conclusion or interpretation of the results. They are only compared to the control run and not to independent data sources; to me it is not clear how this results in a validation of the parameter choices in the control run.

Following the responses from your general remarks 2, we considered that the results of control run are the optimal one (an analytic expression of WISSDOM scheme). In this study, the most important works are to evaluate the impacts from each observation that is the reason why we designed a series sensitivity testing. The results of sensitivity testing present the potential range of errors from each independent observation/input. We also try to include any kind of possible parameters, which is affecting the results of control run. It will be helpful to verify the impacts from various scenarios in this area. The conclusions can also be good refence to decide where the best locations for the instruments employed. We have further emphasized these statements in the beginning of Section 5 and in the Conclusion section as:

L559-560. In this session, the impacts of various datasets implemented in the WISSDOM synthesis were evaluated, then the range of errors can be estimated from

The algorithm looks like an elegant solution to merging multiple data sources into one cohesive 3-D wind dataset. However, the verification and sensitivity tests are not quite convincing yet. The discussion needs a clearer interpretation of the results and it would be nice to compare the control run to the initial version of WISSDOM.

One of goals in this study is that the algorithm was designed in more flexible way to include more useful inputs as possible. As our responses of point 2 in your general remarks 1, the Doppler lidar and Doppler radar cannot co-work at the same time. Usually, the Doppler lidar is operated under clear-air condition only. Thus, it is not easy to derive the winds if we want to implant the Doppler radar data in old version of WISSDOM under clear-air conditions. However, we can implant the Doppler lidar data in a new designed test of Experiment A (i.e., A-5). Basically, the A-5 is likely equaled to the old version of WISSDOM except for the input data (we used lidar here), therefore, its results can be good reference to understand the ranges of error from the control run (i.e., discrepancies between control run and old version of WISSDOM). The contexts related to the verification with independent observations and the interpretations on sensitivity testing have been added clearly in the manuscript following previous responses on your first 3 general remarks. Also, we have putted the discussions and modified figures associated with the results of A-5 in the revised manuscript as well (please refer to our response point 4 in your general remarks 1).

**Detailed remarks:**

A line-by-line review is in the attached, annotated PDF.

Please also note the supplement to this comment:

https://amt.copernicus.org/preprints/amt-2022-218/amt-2022-218-RC1-supplement.pdf

Revisions and response to the line-by-line reviews: We have carefully checked the comments in attached PDF, please see the details from our responses as below:

L1. if WISSDOM existed before, is it not rather improved than newly developed?

Thank you for this valid point. The words "newly developed WISSDOM" have been corrected to "modified WISSDOM" throughout the manuscript.

L42. In the main body of the manuscript, it would be helpful to have a graphical summary of all error metrics.

The schematic diagrams and the descriptions on the summaries of these results have been added in the Conclusion section (Fig. 18).

L48-51. How? you compare the sensitivity test to the control run, not an independent reference.

We have emphasized that the derived winds from the control run are the optimal results (i.e., the analytic expression of variational-based scheme in WISSDOM) in the texts. Thus, the descriptions of this sentence have been modified for clearly as:

L36-39. A series of sensitivity tests with different weighting coefficients, radius of influence (RI) in interpolation and various combination of different datasets were also performed, and the results indicate that the present setting of the control run is the optimal reference to WISSDOM synthesis in this event.

L71. definition missing.

The full name of EGBVTD (Extended-Ground-Based Velocity Track Display) has been added in revision (L60-61).

L78-79. Acronyms are not defined consistently throughout the text -> sometimes the full name is first, sometimes the short version.

All of acronyms have been defined consistently throughout the manuscript (full name is first).

L84-87. relevance? The manuscript overall is very long. Removing information that is not crucial to the understanding would help to keep the focus on the relevant points.

The redundant information have been removed.

L89-91. these are the main references for the previous method. please describe briefly, which one contains what.
it can also be helpful to mention this again in the methods section and clarify which parts of the methods were described in which of these sources.

We have clarified the importance and contains for each reference in the sections Introduction and Methodology as below:

L77-80. The first purpose and details of algorithms can be found in Liou and Chang (2009). Performing immersed boundary method (IBM, Tseng and Ferziger, 2003) in WISSDOM and its scientific applications were clearly documented in Liou et al. (2012), and Liou et al. (2016), respectively.

L135-144. The original version of WISSDOM performed five constraints, including radar observations (i.e., reflectivity and radial velocity), background (combined with

automatic weather stations, sounding, model or reanalysis data), continuity equation, vorticity equation, and Laplacian smoothing (Liou and Chang 2009). Liou et al. (2012) applied the IBM in WISSDOM to consider the affecting upon nonflat surface, one of advantages in IBM is providing realistic topographic forcing without the need to change the Cartesian coordinate system into a terrain-following coordinate system. More scientific documentations associated with the interaction between terrain, precipitation and winds in different areas can be referred to Liou et al. (2016, Taiwan), and Tsai et al. (2018, South Korea).

L91. Remove "synthesis".

The redundant word has been removed.

L92. I am not able to find this part in the methods section, can you point it out more clearly?

The descriptions about the main benefit on the IBM have been added in the methods section as below:

L139-142. Liou et al. (2012) applied the IBM in WISSDOM to consider the affecting upon nonflat surface, one of advantages in IBM is providing realistic topographic forcing without the need to change the Cartesian coordinate system into a terrain-following coordinate system.

L107. with multi-Doppler analysis?

Yes, the sentence has been modified for clearly.

L94-96. Bell et al. (2020) combined an intersecting range height indicator (RHI) of six Doppler lidars to build "virtual towers" (such as wind profilers) to investigate the airflow over complex terrain during the Perdigäo experiment.

L108-109. what is a conventional physical tower?

It indicates the real meteorological tower here; the sentence has been modified for clearly.

L96-97. These virtual towers can fill the gap in wind measurements above meteorological towers.

L113. what is the reference for the original WISSDOM? Since it originally was with Doppler radars, did it contain dealiasing?

The main reference of original is Liou and Chang (2009), and it has been indicated in L77-78: The first purpose and details of algorithms can be found in Liou and Chang (2009). Besides, since we performed the variational-based method to get the derived winds with an analytic expression, the effects on the dealiasing should be mitigated

L114. There is not a great deal of discussion, whether changing from radar to lidar data requires a different data treatment approach. E.g. the advantages of LIDAR in clear air are highlighted, but how would you handle attenuation in precipitating conditions?

The Doppler radar data was used in the original design of WISSDOM. However, we used the Doppler lidar data in modified WISSDOM under clear-air condition. The main objective of this study is to appropriately implant the Doppler lidar data in modified WISSDOM and included more observational inputs as possible. It is not necessary to consider the issue like changing radar to lidar in modified WISSDOM scheme in this stage. Thus, we don't have to handle the problems like attenuation in precipitating conditions since only Doppler lidar data were adopted in WISSDOM under clear-air condition in this study. In the future, the ambiguous conditions between clear-air and precipitation may have further discussions to see how to implant or switch the inputs from lidar and radar data. We have emphasized the uses of lidar data in revise manuscript, please refer to our responses point 2 in your general remark 1.

L128-129. how was the spatial resolution chosen?

Since the spatial resolution of lidars are 40 m and 60 m at the DGW and MHS sites, respectively. We preferred to apply an optimal value (50 m) in modified WISSOM based on the main inputs from these two lidars. This information has been added in a sentence as:

L117-120. This modified WISSDOM will allow us to obtain an exceedingly high spatial resolution of 3D winds (50 m was set in this study) under clear-air conditions. It is because the Doppler lidar had high spatial resolution in between 40 and 60 m horizontally…

L139. With WISSDOM being an established algorithm, you can benefit from existing descriptions of the algorithm and shorten this section. Summarizing it in a schematic diagram can be a concise way to explain the previous setup.

Since we have made sufficient descriptions about the methods and their corresponding references in this section. We preferred to keep most parts of texts in this section to more clarify the context from original to modified WISSDOM, and the reason why we used Doppler lidar data but not uses Doppler radar data. It is also helpful for that the algorithms can be reproduced from the information given.

L152. is the time constraint only used in the Doppler data?

No, the time constraint is not just only used in Doppler data, but also used for all constraints. The descriptions have been corrected as:

L152-154. Since WISSDOM is a scheme that uses the 4DVAR approach, the

variations between different time steps ($t$) should be considered, and two time steps of radar observations were collected in this constraint and all following constraints.

L197-199. Is there a methodology change related to this, or is it just changing the input data?

No, it is not just only replaced the data by lidar observations, but also relatively reasonable assumption should necessarily applied here. Since the terminal velocity of precipitation particles can be neglected under clear-air condition, $W_{T,t}$ was set to zero in eq. (3) in the modified WISSDOM. The descriptions have been modified doe clearly in the texts as:

L200-202. Instead of the radial velocity $(V_r)_{i,t}$ observed from Doppler radars in eq. (3) in original version of WISSDOM, the radial velocity observed from Doppler lidars was adopted in the modified WISSDOM synthesis. In addition, if there were no precipitation particles under clear-air conditions, the terminal velocity of precipitation particles ($W_{T,t}$) was set to zero in eq. (3) in the modified WISSDOM.

L227-229. So it was used before the method was verified? To what extent does this publication offer additional value to Tsai 2022?

Tsai et al. (2022) uses the results of control run from modified WISSDOM to investigate more scientific issue related to orographic winds. The conclusions helped us to more understand the possible role of terrain on modifications of winds in the area. In different view standing from our submission in AMT (i.e., this article), it provides clear context, detail procedures, reliability, and the limitations of the modified WISSDOM. The conclusion from this study will benefits the community who want to use WISSDOM in clear-are conditions. These statements have been added in the texts as:

L236-238. In different view standing from previous studies, this study provides clear context, detail procedures, reliability, and the limitations of the modified WISSDOM.

L229. The actual process of obtaining the final wind values is still unclear. Is there a first guess and a gradient descent technique to find the minimal loss? Is the first guess random, or where does it come from? Ideally this section should be detailed enough, that the algorithm can be reproduced.

This is valid point. Yes, there is a first guess random and a gradient decent technique to find the minimal loss in WISSDOM. The first guess is usually coming from sounding, modeling, however, it can be zero if there is not any information in beginning. The descriptions about this issue have been added in the revision for clearly as:

L148-150. Note that the $\mathbf{V}_t$ will be first guessed random resulting from sounding, modeling, or equal zero if there is not any information about wind in beginning.

L237-239. How exactly was this chosen, what are recommended limitations?

It mainly depend on the scale of instating weather systems, and temporal resolution of lidars (10-20 mins). Usually,10-60 mins are acceptable interval for the analyses in micro to mesoscale phenomena. We also have remarked the recommended limitation is 10 mins in the text as:

L246-248. Note that the output time steps are adjustable to be finer (recommended limitation is 10 mins), but they are highly related to the temporal resolution of various datasets and computing resources.

L240. acronym not defined.

The acronym has been defined [May Hills Supersite (MHS)].

L421. acronym not defined.

The acronym has been defined [DaeGwallyeong regional Weather office (DGW)].

L247. The differentiation between the different symbols rather difficult in this plot. Especially the square under the star is not well visible. Consider using different colours.

The Figure 1 has been modified for clearly based on the comments. Please find the revised figure as below.

[Figure]

L249. this is rather a star shape than an asterisk.

The captions has been corrected to start symbols in the text.

L250. Is there more than one square?

No, the word has been corrected to "square" in the caption.

L258. A full explanation of a volume scan, PPI and RHI is very long. I would recommend to only define the acronyms and ommit the explanation, these are standard terms in the remote sensing community.

The redundant descriptions have been removed.

L259-260. likely? this should be confirmed.

This sentence has been removed.

L281-282. How well does this work for wind in particular? Wind fields are notoriously difficult to interpolate.

The wind direction and speed must first project with the values along u- and v-components, then interpolate these values to given grips (i.e., Cartesian coordinate system) by adopting objective analysis. The descriptions have been added in revision as:

L286-288. Note that the wind directions and wind speed must first project with the values along u- and v-components then interpolate their values individually to the given grids.

L343-345. If all data sources were interpolated to the analysis grid, how was the analysis grid chosen?

The Cartesian coordinate is the most efficient way and the best system for partial differential equation (Armijo, 1969), then it also be used in the cost function of WISSDOM (Liou and Chang, 2009). Furthermore, the IBM has the merit of providing realistic topographic forcing without the need to change the Cartesian grid configuration into a terrain-following coordinate system (Liou et al. 2012). The descriptions have been cleared in the revision as:

L348-350. The Cartesian coordinate is the most efficient way and the best system for partial differential equation (Armijo, 1969), then it also be used in the cost function of WISSDOM (Liou and Chang, 2009).

L371-372. Is this given by the scan strategy or were there circumstances reducing the lidar coverage?

It was caused by the scan strategy due to one of main purposes (make sure the safety of athletes and audiences) during the Olympic games. This sentence has been modified for clearly:

L377-379. The coverage of lidars was reduced significantly above 900 m MSL and remained at ~5% due to the scan strategy during the Olympic games (more dense observations near the surface).

L385-386. How were the weights and AWS interpolation of the control run chosen? Why is the control run more reliable than those experiments?

The locations of the AWS stations are presenting more random distributions with the distances in 100 to 2000 m between each site in study domain (cf. Fig. 1). In addition, since the best weights have been determined by a series of observation system simulation experiment (OSSE) type tests from Liou and Chang (2009), they have putted more weights in observations and less weights in modeling inputs. Therefore, we first considered the best setting about the AWS are shown in table 1

based on the default weights from previous studies and the real conditions in our study area. According to our responses point 3 to your general remark 1, the outputs of WISSDOM are the optimal results when it appropriately included all available wind information as possible. Thus, we have first compared the discrepancies of results between the control run and the observations, Finally, we have designed several experiments to understand the limitations of modified WISSDOM. In this case, the control run presented good performance, and the conclusion from designed experiments indicates that unreasonable values were explored if the inputs are not appropriately included. Related descriptions have been modified for clearly as:

L392-393. Relatively reliable 3D winds were derived by a control run of the WISSDOM synthesis because all available wind observations and local reanalysis datasets were appropriately acquired.

L395-397. Therefore, the retrieved winds from the control run can be treated as the optimal results (i.e., analytic expression of variational-based scheme) in WISSDOM.

L403-407. Note that the best weights have been determined by a series of observation system simulation experiment (OSSE) type tests from Liou and Chang (2009), they have putted more weights in observations and less weights in modeling inputs. Based on the experiences and the default setting of weights from previous studies, the basic setting of the control run has been first decided.

L388-389. what are these?

The necessary procedures in data implementation indicates the way how to implant all lidar, AWS, sounding, and LDAPS data in WISSDOM. Those procedures can be found in the sentence as follows. This sentence has been modified for clearly as:

L397-398. The control run was performed carefully with the necessary procedures in data implementation before running the WISSDOM synthesis as follows.

L392-393. there are more/different inputs than in the original WISSDOM version - how were the weights chosen here?

The default setting of weights for Doppler radar data is $10^6$, and for the background is $10^2$ (Liou and Chang, 2009). There were more heavy weights in observational data and relatively less weights in background (they used modeling in background). In this study, the weights of additional constraints were basically set following their concepts. The descriptions have been added the same as last two responses.

L395. Why did you choose this the way it is?

We have considered the real situations/conditions in our study domain like the distributions of available datasets, it allow us to perform the best setting of the control run. In addition, we also followed the default setting of weights and the concepts as more weights in observations and less weights for the modeling inputs. Please refer to our above three responses, we have explained how to decide the

weights and the descriptions about these have been also clarified in the texts.

L416. u and v should be on the same colorscale to make interpretation more intuitive.

Thank you for pointing out the problem. The color scale has been modified in Figure 5 for intuitive as below:

[Figure]

L427-428. This paper describes quasi vertical profiles for polarimetric radar variables. For velocity profiles it suggests a combination of VAD and vertical scans. It is not clear how the lidar QVPs were obtained from this description alone. Is one profile generated per LIDAR or are both LIDARS merged?

A profile of the QVP was generated from a LIDAR. Thus, the descriptions have been added in revision for clearly as:

L439-441. Two kinds of instruments were available in the test domain to detect the relatively realistic winds: sounding and lidar quasi-vertical profiles (QVP, Ryzhkov et al., 2016), a profile of QVP can be general form a lidar.

L443-444. to what extent is the time of measurement in the sounding relevant? It takes time to obtain a full profile.

Dense PPI at lower layer provided good quality of observations at lower layers. Although a full scan is taking long time, it only takes ~6 mins to finish the scan below 3 km MSL (i.e., the top boundary of this study). The sounding is taking ~7 mins to reach 3 km MSL. Thus, we considered that these two datasets are comparable.

L466-467. it is unclear if Fig. 8 discusses the difference of wind direction, or the absolute value.

The difference of wind direction was discussed here, the sentence has been modified for clearly as:

L479-482. Except for relatively larger IQR existed (between ~−5 and 5 degrees) and larger median values (between ~0 and 5 degrees) can be found at the lowest level, the interquartile range (IQR) and median values of the wind direction differences are smaller (between ~0 and 2.5 degrees) during the entire research period (Fig. 8a).

L483-484. Fig. 8b shows the difference in wind speed between the WISSDOM synthesis and sounding observations.

L486-487. The median values of the wind speed differences were between −1 and 0.5 m s$^{-1}$, and the IQR of wind speed differences was between −2 and 0.5 m s$^{-1}$.

L467. sentence unclear.

This sentence has been modified for clearly as:

L479-482. Except for relatively larger IQR existed (between ~−5 and 5 degrees) and larger median values (between ~0 and 5 degrees) can be found at the lowest level, the interquartile range (IQR) and median values of the wind direction differences are smaller (between ~0 and 2.5 degrees) during the entire research period (Fig. 8a).

L482-483. Does this merge the information from the second LIDAR? Is the uncertainty of LIDAR QVP inherently higher, if u and v are estimated from a VAD procedure, averaging a larger volume?

No, it does not merge the information from the other lidar here. The uncertainty of Lidar QVP is more related to the coverage of lidar observations at different layers (relatively dense at lower layers, cf. Fig. 4). Therefore, better retrieved winds are more concentrated at lower layers but not at higher layers.

L551-552. This sentence does not make sense.

This sentence has been modified for clearly as:

L560-562. The basic setting of Experiment A took off several inputs from the WISSDOM control run (cf. Table 1) as four designs in Experiment A.

L564. you state earlier that this study also uses WISSDOM - how is this an independent verification?

Simple verifications were applied qualitatively by checking the winds of ERA5 reanalysis data in Tsai et al., (2022). That is also a reason why we should check the performance of modified WISSDOM in detail via present study.

L570. Is it possible to perform a run with the "old" WISSDOM version (though keeping LIDAR instead of Doppler data), where soundings, AWS and LDAPS are merged into the background?

A new design has been established to understand the possible discrepancies between old and modified versions of WISSDOM. Please refer to our responses point 4 to your general remarks 1 in detail.

L570. what about the weights of the soundings or background?

The weights of sounding and background have been added in Table 4.

L572. it somewhat sounds like the AWS worsen the quality, please rephrase to clarify that the windspeed is reduced.

The descriptions have been modified perorally as:

L585-586. The impacts of the AWS cause negative values on the u-component winds in most areas at 800 m MSL in A-2 (Fig. 12c), especially in the western areas of the MHS site.

L603. why? how do you know it is insufficient?

Since the altitude of the AWS at the MHS site is relatively low compared with the other AWS nearby. Extending the VE can sufficiently include more data not just only from the MHS site but also from the other AWS sites. The results also reveals that this unusual circle can be vanished when VE becoming 90%. The sentence has been modified for clearly as:

L628-630. An unusual circular area with positive discrepancies around the MHS site was depicted in B-1 (Figs, 14a and 14b), which may have been produced by the insufficient RI distance and VE (unusual circle can be vanished when VE becoming 90%).

L605. Remove "AWS stations".

The sentence has been rewritten for clearly as:

L631-632. Relatively smaller RI and VE values can only include relatively less wind information if the distances are large between each AWS station.

L613. The acronym AWS is used inconsistently throughout - sometimes the plural is denoted with AWSs, sometimes not, sometimes the word station still appears redundantly

Please revise throughout the manuscript.

It has been modified throughout the manuscript.

L634. The conclusion from this is not clear. How do you know the control run settings are the best? Would you like to show that the results are robust to changes in the interpolation distances?

According to our responses point 3 to your general remarks1 as the derived winds of control run is the optimal result. And the conclusions have been added for clearly. We have checked the interpolation distances from 0.5 to 2 km, and results reveals robust changes (cf. Fig. 15)

L661-662. The conclusions indicated that the moderate setting (i.e., RI is 1 km) will be helpful to get the smallest differences with the control run.

L641-642. why, where do these come from? the absolute values seem arbitrary, does this relate to the number of datapoints in each dataset? is only the relative difference between the weights important?

The weights were decided considering the default setting in previous study (Liou and Chang, 2009). They have figured out the optimal weights for each input by a series test with the OSSE. Furthermore, we consider that it does not link to datapoint too much.
No, it also has no direct relations with the elative difference between the weights important. Basically, higher (lower) weights can be determined in observational (modeling/reanalysis) inputs from the OSSE tests.

L646-647. How come these two data sources require much higher weights than the LDAPS, is it because their grid is less dense?

Following the responses above: it was because the default weights determining from the OSSE tests in Liou and Chang (2009).

L648-649. do you have a hypothesis, why this happens?

We had a hypothesis, but it does not confirm yet; we consider that the results cannot converged well due to huge values with low spatial resolution from the LDAPS

compared with the lidar data. It looks that the weights of LDAPS may be too high to cause these results. Since it does not confirm yet, we prefer to keep this hypothesis here, but not showing in the texts.

L663. Also here, it is unclear what your conclusion is from this. Robustness to changes in the weighting coefficients? But then, is it really robust, if changing the weights of LDAPS has such a strong effect?

The results reveals that the changes of the weights are too sensitive to the derived winds, especially for the inputs of the LDAPS and AWS datasets. Therefore, the weights of LDAPS and AWS are not necessary changed too much.

L694-696. The conclusions reveals that the weights of the AWS and LDAPS (lidar) are (not) too sensitive to the derived winds. Therefore, the weights of LDAPS and AWS are not necessary changed too much.

L664. what happens here? is this the sounding location?

Yes, this is the track of sounding, please refer to the figure 2s in left as below.

[Figure]

L682. it would also be helpful to provide some context, which bias/RMSD is acceptable and what is too large.

We have provided the context and the performance of modified WISSDOM compared with the conclusions from Chen (2019) in second paragraph in Conclusion section.

L707-708. So what do you conclude from this?

The results indicates that the optimal setting of the RI is 1 km. The descriptions have been modified for clearly.

L747-749. In Experiment B, the smallest discrepancies in 3D winds were depicted when the RI (VE) was set to 1 km (50%); it indicated that the optimal setting of the RI is 1 km.

L708-712. Also here, what should the reader take away from this?

The most key conclusion can be found as below, and we have added these descriptions in the last paragraph in Conclusion section as well.

Reasonable winds can be retrieved from modified WISSDOM with sufficient coverage from the data, moderate weighting function and appropriate implements from different datasets.

L723-725. how was the wind profiler data used?

The words have been removed from here.

Reference

Armijo, L.: A theory for the determination of wind and precipitation velocities with Doppler radars. J. Atmos. Sci., 26, 570–573. 1969.

Liou, Y., and Chang, Y.: A Variational Multiple–Doppler Radar Three-Dimensional Wind Synthesis Method and Its Impacts on Thermodynamic Retrieval. Mon. Wea. Rev., 137, 3992–4010, https://doi.org/10.1175/2009MWR2980.1, 2009.

Liou, Y., Chang, S., and Sun, J.: An Application of the Immersed Boundary Method for Recovering the Three-Dimensional Wind Fields over Complex Terrain Using Multiple-Doppler Radar Data. *Mon. Wea. Rev.*, **140**, 1603–1619, https://doi.org/10.1175/MWR-D-11-00151.1, 2012.

**General comments:**

This study presents the advanced WISSDOM synthesis scheme that can provide high- resolution three-dimensional wind fields in clear weather, which was only available in storms before. The authors examined the performance of the wind retrieval from the advanced WISSDOM in a strong wind case over complex terrain. The mean biases are generally small, demonstrating that the new WISSDOM is capable of providing reliable wind fields in fair weathers and topography. Several sensitivity tests are also performed to determine the optimal settings of the new WISSDOM system.

The manuscript is organized and well-written. The figures are well illustrated. Yet, there are some issues that need to be clarified and addressed before publication.

We appreciate Referee#2 providing helpful and insightful comments, which help us to improve the manuscript substantially. A set of responses to your comments is provided below. Specific locations of modified portions (marked as underlines) were also noted as the number of lines in the revised manuscript.

**Specific comments**

1. The authors quantify the sensitivity tests and discuss these tests respectively in table 2 and Figs. 6 and 9. Yet, these only state the number of the biases but without further discussions. Having some more discussion and a summarized recommended recipe in conclusion would be more helpful for the future users, e.g. what observations are the most critical in some cases, how to determine the related RI/VE, and why these suggestions may work (e.g. the height of the PBL, decorrelation of the atmospheric state based on observations, etc.).

Thank you for this valid point. Significant discrepancies can be found without some inputs like the AWS and Lidars. More discussions about the importance of each input have been added in the last paragraphs in 5.1. Please check these parts as below:

L611-616. The wind speed can be better modulated in modified version of WISSDOM when the Doppler lidar observations were adopted. These results also implied that the ranges of errors are relatively small when we try to evaluate the discrepancies between the control run and each independent observation. In summary, the results of this experiment (cf. Fig. 13) concluded that the lidar and AWS data are more critical inputs in modified WISSDOM, and it will be benefits if more inputs can be included.

Since the average distance between each AWS site is from ~100 to 2000 m, and the main purpose of the modified WISSDOM is trying to include more and complete data. Therefore, the setting of this sensitivity testing is from 0.5 to 2 km in horizontally and from the default setting (50%) to extending setting (90%) in vertically. The conclusions, main reasons how to determine the setting of RI/VE, and more

discussions have been modified for clearly as:

L623-626. There were five designs (B-1~B-5) in Experiment B with the ranges of RI (VE) between 0.5 km (50%) and 2 km (90%). Because the average distance between each AWS site is approximate from 0.1 to 2 km and more data can be included in vertically.

L661-665. The conclusions indicated that the moderate setting (i.e., RI is 1 km) will be helpful to get the smallest differences with the control run. In addition, the wind directions and speed should be more dominated by terrains at lower layers, the implements of AWS data are very important for the modified WISSDOM synthesis, especially at the height below 900 m MSL.

- Why is the average bias higher between WISSDOM-lidarQVP than between WISSDOM- sounding?

  First, the sounding data have been adopted in WISSDON following its tracks; second, the QVP were made via more dense observations at lower layer but not higher layer. Therefore, the results of WISSDOM synthesis are in good agreement with the sounding for each level and with QVP only at lower layers. The descriptions and explanations have been emphasized for clearly as:

  L210-211. The sounding data in $I_6$ were interpolated to the given grid points near its tracks bearing on the radius influence (RI) distance (the details are provided in Section 3.2.3).

  L448-449. Fig. 6 shows the scatter plots of the u- and v-component winds on the locations following the tracks of sounding launched from the DGW site.

  L528-532. In summary, the discrepancies in the 3D winds between the WISSDOM synthesis and lidar QVP were small in the lower layers and large in the higher layers because the observational data from lidars and AWS provided good quality and sufficient wind information at the lower layers but not in the higher layers (lower coverages of lidar data above 1.3 km MSL, cf. Fig. 4).

- Why is v-component correlation of WISSDOM-lidarQVP particularly lower (down to 0.38)?

  Following our responses above, relatively less data from lidar QVP were adopted at higher layers. We have calculated the correlation coefficients below 2 km height MSL, and the results shows higher correlation (~0.45). The descriptions have been added in the texts as:

  L501-504. The correlation coefficient of v-component winds is lower (0.35) in association with low wind speed (<15 m s$^{-1}$) from the surface to 2.5 km MSL (Fig. 9b), and it may possibly relate to less coverages from lidar QVP data at higher layers.

- In addition, what sensitivity tests do the authors suggest readers must do based on your experiences if they are interested in applying the WISSDOM system in different terrains and the weather systems (e.g. typhoons, land-sea breeze etc.)?

In our case, reasonable winds can be derived over complex terrain when the VE was set as 90%. We would like to recommend that 90% (50%) of VE should works well over complex terrain (flat surface) in modified WISSDOM. Lidar is good tool to derive the winds, it will be useful to apply the modified WISSDOM in the analyses of land-sea breeze, low-pressure system without precipitation, micro-downburst, and any meteorological phenomena under clear-air conditions. The descriptions have been added in the manuscript as:

L752-754. In addition to the reasonable winds can be derived by applying the optimal setting in modified WISSDOM, 90% (50%) of VE are also recommended over complex terrain (flat surface).

L767-769. Except for the detailed wind structures can be well documented in any meteorological phenomena under clear-air conditions (eq., land-sea breeze, micro-downburst, and non-precipitation low-pressure systems etc.).

2. Sec. 3.2.1 and 3.2.2 Why use the interpolated observation data in the Cartesian coordinate instead of the original high-resolution observations of scanning Doppler lidars and AWS? Why use the lidar QVP instead of the high resolution lidar information?

Usually, the most efficient way to minimize the cost function is to use Cartesian coordinate for partial differential equation (Armijo, 1969). Besides, the default setting of original WISSDOM were performed on Cartesian coordinate as well (Liou and Chang, 2009). Thus, we performed the variational calculations on the Cartesian coordinate in modified WISSDOM in this study. Based on this, we were finally interpolated any input data (including the lidars and AWS) to Cartesian coordinate here. The descriptions about these statements have been added in the revision clearly as:

L268-273. Although relatively dense and complete coverage of wind information (i.e., radial velocity of aerosols) were sufficiently recorded by lidar observations, the collected data are usually not located directly on the given grid points in the WISSDOM synthesis (i.e., Cartesian coordinate system). In this study, the lidar data were interpreted simply from the lidar coordinate system to the Cartesian coordinate system via bilinear interpolation.

L348-350. The Cartesian coordinate is the most efficient way and the best system for partial differential equation (Armijo, 1969), then it also be used in the cost function of WISSDOM (Liou and Chang, 2009).

3. In the evaluation of the control run, the observations used for evaluation are also used in the wind retrieval by WISDOM. Are there other independent observations that can be used for evaluations?

Since the basic idea of WISSDOM is variational-based scheme to minimize every constraint in cost function. The optimal results can be calculated when it included more data as possible. According to our responses to the same comments from Refee#1. We have emphasized that the retrieved winds from the control run are the optimal results (i.e., the analytic expression of variational-based scheme in

WISSDOM) in the texts as below:

L133-135. The 3D winds were derived by variationally adjusted solutions to satisfy the constraints in the cost function, thus the results of retrieved winds were the analytic expression in this scheme.

L395-397. Therefore, the retrieved winds from the control run can be treated as the optimal results (i.e., analytic expression of variational-based scheme) in WISSDOM.

4. Sec. 2.1: For the methodology:

- In fair weather, how are the radar reflectivity and radial velocity constrained in new WISSDOM? Do you use clear echoes or insects? Do you use both Z and Vr?

  In modified WISSDOM, only lidar radial velocity were used because there was no reflectivity can be detected by radar under clear-air conditions. In addition, we did not use any echoes and insects in modified WISSDOM since we assumed the terminal velocity of air mass are small and can be neglected. Thus, the terminal velocity of particle can be set to zero in eq. (3). The description can be found in:

  L200-204. Instead of the radial velocity $(V_r)_{i,t}$ observed from Doppler radars in eq. (3) in original version of WISSDOM, the radial velocity observed from Doppler lidars was adopted in the modified WISSDOM synthesis. In addition, if there were no precipitation particles under clear-air conditions, the terminal velocity of precipitation particles $(W_{T,t})$ was set to zero in eq. (3) in the modified WISSDOM.

- Why do the authors include the vorticity equation in the cost function? What are the advantages and disadvantages?

  The vorticity equation is not first applied in wind retrieval scheme, and Liou and Chang (2009) have also applied the vorticity equation in original version of WISSDOM. As the conclusions from above study, the main advantage is that the use of the vertical vorticity equation can provide further improvements to the winds and thermodynamic retrievals. The descriptions have been added in the text as:

  L190-192. Note that the main advantage is that the use of vertical vorticity can provide further improvement in winds and thermodynamic retrievals.

- What are the time steps in the 4DVAR system of WISSDOM?

  We used 12 mins to be the interval between two time steps in our modified WISSDOM. It is because the temporal resolution of main input lidar data is 12 mins. The descriptions about these have been added in the text for clearly as:

  L204-206. The time steps in WISSDOM of this study were set by the synthesis time and 12 mins before the synthesis time due to the temporal resolution of main input lidar data is 12 mins.

**Minor comments:**

1. L 60: It will be helpful to specify what weather systems.

The specific weather systems have been indicated in the text in:

L46-50. Most comprehensive applications of the derived winds were adopted to document kinematic and precipitation structures associated with various weather systems at different scales like typhoon, tropical cyclone rainband, and non-precipitation low-pressure system (LPS) (Yu and Tsai, 2013, Yu and Tsai, 2017, Tsai et al. 2018, Yu et al., 2020, Cha and Bell, 2021, Tsai et al., 2022).

2. L 67-77: I'm not sure how the VTD/GBVTD/EGBVTD are related to the developing of WISSDOM system or three-dimensional wind retrieval. It's not clear. Please clarify it.

The reason why we mentioned those schemes in advance, because we are trying to emphasize the advantages of WISSDOM. The descriptions about the contexts and explanations have been added for clearly as:

L62-67. However, winds usually present nonuniform patterns and fast-evolving characteristics in most mesoscale weather systems and microscale phenomena, and complete and detailed winds are still difficult to resolve by these techniques. Based on the contexts of weaknesses from above schemes on the wind retrievals. Instead of a single Doppler radar, multiple Doppler can retrieve better quality 3D winds with relativity fewer assumptions because they provide sufficient radial velocity measurements and wind information with wider coverage in the synthesis domain.

3. Figure 16f: There is a very artificial band in Fig. 16f, the difference between the control run and the C3. Please explain the artificial pattern.

These artificial band (showing in the right figure below) are highly related to the sounding since its locations are almost the same with the track of raising sounding in this study (as Fig. 2a, thick black line, left figure below).

[Figure]

Technical issues:

1. L 153: The acronym of the time steps (t) and two time (t) steps are the same. It's confusing.

The acronym has been removed for clearly.

2. L 290: gird → grid

Thanks for indicating the typo, and this typo has been corrected.

3. Figure 3. It will be helpful to indicate the location of the analyzed area on the synoptic surface map for readers who are not familiar with Asia.

The new figure has been replaced to emphasize the location of the Korean peninsula (as below), and the descriptions have been also added in figure captions.

[Figure]

Figure 3. Synoptic surface chart from the Korea Meteorological Administration (KMA) at 00:00 UTC on 14 Feb. 2018. The locations of the Korean peninsula and the LPS has been marked by black circle.

Reference

Armijo, L.: A theory for the determination of wind and precipitation velocities with Doppler radars. J. Atmos. Sci., 26, 570–573. 1969.

Liou, Y., and Chang, Y.: A Variational Multiple–Doppler Radar Three-Dimensional Wind Synthesis Method and Its Impacts on Thermodynamic Retrieval. Mon. Wea. Rev., 137, 3992–4010, https://doi.org/10.1175/2009MWR2980.1, 2009.

---

## Referee Report (RR1)

**Review** "High Resolution 3D Winds Derived from a Newly Developed WISSDOM Synthesis Scheme using Multiple Doppler Lidars and Observations"**

Thank you for taking into consideration the suggestions made in the first round of revisions. The study benefits from the additional experiment and the added information clarifies the WISSDOM procedure and the parameter choices. Nonetheless there are still a few points that remain unclear after the revisions.

**General remarks:**

Overall, the level of English has worsened in the new text additions. Some sections are highlighted in the specific remarks below, but overall a language revision by a native speaker is strongly advised.

The additions in the algorithm section are very helpful, however a few questions remain. You mention using soundings for the background and as a separate constraint, how does J2 differ from J6 in the modified WISSDOM? Why do you not include all possible sources in the background? Please also mention the use of gradient descent to converge to a solution explicitly in the text.

The procedure on how to retrieve the LIDAR QVPs is still not clear to me. Could you provide a small paragraph describing the method? Do you use a VAD technique to estimate the vertical profile of horizontal wind, as suggested in Ryzhkov 2016? And if yes, how do you estimate vertical wind without having a vertical scan? Do you derive a QVP over each LIDAR or just the DGW site as suggested in Fig. 10? Please expand the text accordingly.

In Fig. 16, both reviewers pointed out the artificial band resulting from the sounding, please include an explanation in the text, noting why this leads to such an artificial anomaly.

From the additions in the text, it is still not clear, why the control run is objectively the best run, especially with regard to the sensitivity tests in experiments B and C. While the control run itself is compared to measurements – albeit not independent ones – the experiments A, B and C are only compared to the control run. This does not allow us to see, whether this difference actually results in an improved or worsened performance with respect to the measurements. Hence it also does not provide the information necessary to determine that the control run performs better. If the goal is to show that the control run is optimal, then comparing all experiments to the measurements would make more sense. Since this modified version of WISSDOM includes more

input data sources (i.e. more constraints in the cost function), it is valuable to reassess, whether the choice of weights (experiments C) still performs well. The parameter choice for the integration of the AWS data (experiments B) seems to be highly dependent on AWS location and topography and hence also requires a verification against measurements here.

In conclusion, I would suggest to show the results of the sensitivity tests in experiments A, B and C with respect to the sounding and the QVP and adapt the discussion accordingly.

With respect to the verification measurements being used in the algorithm, it is important to state clearly in the text that these are not independent measurements and that the control run is not verified independently. The further experiments A can then show, how WISSDOM performs against soundings, when soundings are not used (as an example), or against QVP, if LIDAR is not used.

**Specific remarks:**

The following line references highlight minor issues. The line numbers refer to the revised, markedup manuscript. Line 32: automatic weather stations (AWS) – throughout the manuscript it is often written AWS station, which is redundant. Please remove "station" after the acronym throughout.

Line 62: It would be interesting to add the characteristic scales to the examples of weahter systems.

Line 78: Based on... - This is a grammatically incomplete sentence.

Line 96: Performing immersed... - Please revise the language.

Line 122ff: This sentence does not make sense grammatically.

Line 138: I would suggest to rephrase to: *A resolution of 50 m was chosen in this study, as the Doppler lidars' respective horizontal resolution averages 40-60m.* The sentence discussing the Doppler radar seems a bit out of place here, what would you like to highlight with this?

Line 159ff: Please revise the language.

Line 168: Please include the description of the gradient descent technique to converge towards a solution. Also, it is not a random guess, if it is based on the available background data, or set to 0, please clarify this in the text.

Line 211: How does WISSDOM perform thermodynamic retrievals?

Line 224: As the sounding and reanalysis data are not available on a 12 min resolution, how is the time constraint performed? You describe that soundings are launched 3-hourly and the reanalysis data is also provided at a 3h timescale.

Line 289: RHI is not defined.

Line 377: My previous question rather addressed the choice of horizontal and vertical resolution, than that of choosing a Cartesian coordinate system. How did you determine your grid parameters? Did you center your gridboxes to the AWS where possible? Is the horizontal and vertical resolution primarily determined by the LIDAR data?

Line 425: I disagree with this statement. The analytical solution for WISSDOM neither prescribes the spatial integration of the AWS nor the weights, from my understanding.

Line 433ff: These weights were determined for Doppler data at a different spatial resolution and for less additional observational constraints. Considering this, the weights should not be seen as optimal per se and reevaluated in experiments C, with respect to the measurements. These sentences also have some grammatical errors.

Line 471: This description of the QVP is still insufficient to understand fully, how it is derived.

Line 509: Please revise the grammar.

Line 535: The lower correlation here may also stem from the QVP method, please elaborate. "Less coverage" instead of "less coverages"

Line 597f: Please revise the phrasing.

Line 605: Please revise the phrasing.

Line 629ff: I do not understand the conclusion here, please rephrase.

Line 663: This sentence is grammatically incomplete.

Line 668: Revise to "the circular artefact is removed when increasing VE to 90 %."

Line 732: The conclusion here is unclear. If the weights are insensitive, does this suggest that the data source does not have a large impact on the outcome?

Fig 16: Please mention the artefact from the sounding explicitly in the text.

Fig 18 b): W for the lidar QVP is not mentioned in the caption.

Line 791ff: Please revise the grammar.

Line 806f: Please revise the grammar.

---

## Referee Report (RR2)

**Review "High Resolution 3D Winds Derived from a Modified WISSDOM Synthesis Scheme using Multiple Doppler Lidars and Observations"**

**General remarks:**

The authors took all of the major suggestions into consideration and added substantial analyses to the evaluation of the algorithm. The additional figures containing the comparison to the observational data now support the conclusions. However, I recommend to add to Figures 13, 15 and 17 b and c the control run as well. It is deemed to be the best performing setting, and this would greatly facilitate the comparison to the other experiments. If the control run is absent from the evaluation, it is still difficult to compare, whether the experiments in A, B and C performed better or worse than the control.

Moreover, the documentation and explanation of the algorithm has improved significantly.

While the language has improved substantially, there are still a number of phrasing issues and the manuscript would benefit from further language editing. Some of these are highlighted in the specific remarks.

**Specific remarks:**

The following line references highlight minor remarks. The line numbers refer to the revised, marked-up manuscript.

General language remarks:

- Many sentences begin with "Note that", in most cases this can be eliminated and the meaning stays the same. Sometimes "note that" has been replaced with "notably", which does not necessarily have the same meaning.
- It is often written that WISSDOM is "performed". This is a rather unusual formulation for the use / execution / running of an algorithm.
- In the description of the results, the tense switches in between past and present.
- The sentences containing many brackets to show many different properties in the same sentence are hard to follow.

Line 16: The WISSDOM (…) synthesis scheme

Line 49: consider associating the scale directly with the phenomenon, e.g. from thousands (cold fronts and low pressure systems) over hundreds (tropical cyclones and typhoons) to a couple of kilometers (convective lines and TC rainbands)
For the TC rainband this naturally depends on whether it is the length or width of the rainband

Line 80: "Liou and Chang is the first purposes of this algorithm." This sentence does not make sense. The following sentence (Furthermore, they performed IBM…) does not read smoothly either.

Line 112: … can be used in the modified WISSDOM.

Line 113: …, which is an essential benefit over Doppler radar data.

Line 134: "In addition, the modified WISSDOM was performed…" performed seems awkward in this context, perhaps execute, ran or used could be alternatives.

Line 136: The reliability of the derived 3D winds was also evaluated and discussed with respect to conventional observations.

Line 145: utilized instead of performed

Line 148: the effect of the non-flat surfaces? The topography itself is not generally affected by the algorithm, on the contrary, it affects the results of the algorithm.

Line 157: Note that Vt is first estimated based on the background of the sounding observations used in this study. In the absence of background observations, the first guess of Vt is set to 0.

Line 188: "In addition, individual constraints…" This sentence is difficult to read, please rephrase.

Line 216: "In this study, the time steps in WISSDOM…" -> In this study the time steps in WISSDOM are set to 12 min, corresponding to the temporal resolution of the primary input lidar data.

Line 304: "Notably, that the wind directions…" This sentence does not make sense.

Line 366: The Cartesian coordinate system

Line 425: They put more weight on observations and less on modeling inputs.

Line 428: variations instead of variances (unless you are referring to statistical variance)

Line 464: Remove "so-called"

Line 471: Do the 2.5 m/s refer to both horizontal and vertical wind deviation? For vertical velocities, this would be a rather large deviation.

Line 514: …relatively larger IQR and median values can only be found at the lowest level…

Line 520-525: Example of switching tenses in the description.

Line 605: An additional test was designed, where only Doppler lidar data are used …

Line 656-664: The distinctions made in the brackets here are rather confusing and hard to follow – it would be easier to read, if differing statements were made in separate sentences.

Line 665: More critical than what?

Line 719: "The conclusions indicated that the moderate setting…" Switching tenses, unclear conclusion -> is this the case with the smallest differences? B3 in blue cannot clearly be identified in Fig. 15. While naturally it is difficult to see all scenarios when they are so similar, it would be helpful to set B3 on top, if you want to highlight it in your discussion.

Line 722: helpless cannot be used in this sense

Line 742: Significant differences often exist in between the observations and reanalysis dataset due to the differing spatio-temporal resolutions.

Line 246: Superposed?

Line 761: The discrepancies in between the derived 3D winds in Experiment C and the sounding observations and QVP, respectively, were also examined.

Line 763: exceeding 20 m/s

Line 764 and 767: This would be easier to follow, if AWS, LDAPS and lidar impact were described in separate sentences.

Line 768: "not necessarily changed much" – in the context of a conclusion, this does not make a lot of sense here, please rather state if and how much they were changed.

Line 779: Emphasize the replacement of radar data and the separation of background wind information.

Line 806: Consider moving the reference to the DGW site to the sentence before. Interquartile range -> IQR

Line 827: raising -> rising

Lines 828ff: Please summarize your final settings in one clear sentence and highlight that they are the same as in the control run.

---

## Author Response (AR2)

**JGR-2022-218 Responses (highlighted in red) to Referee#1 16 Dec. 2022**

Review "High Resolution 3D Winds Derived from a Newly Developed WISSDOM Synthesis Scheme using Multiple Doppler Lidars and Observations"

Thank you for taking into consideration the suggestions made in the first round of revisions. The study benefits from the additional experiment and the added information clarifies the WISSDOM procedure and the parameter choices. Nonetheless there are still a few points that remain unclear after the revisions.

We appreciate Referee#1 providing helpful and insightful comments in this round, which help us to improve the manuscript substantially. We have carefully checked your comments, and more analyses and discussions associated with the performances of the WISSDOM synthesis have been added to this revision. In addition, we put the statements regarding the non-independent evaluations on the results of the control in this manuscript. A set of responses to your comments is provided below. Specific locations of modified portions (marked as underlines) were also noted as the number of lines in the revised manuscript.

**General remarks:**

Overall, the level of English has worsened in the new text additions. Some sections are highlighted in the specific remarks below, but overall a language revision by a native speaker is strongly advised.

We have checked the problematic grammar carefully throughout the manuscript. The native speaker has also revised the texts throughout the manuscript.

The additions in the algorithm section are very helpful, however a few questions remain. You mention using soundings for the background and as a separate

constraint, how does J2 differ from J6 in the modified WISSDOM? Why do you not include all possible sources in the background? Please also mention the use of gradient descent to converge to a solution explicitly in the text.

1. The main difference between J2 and J6 is that the uniform wind speed and wind direction (only one value of u-wind and v-wind) was adopted at each level in the entire WISSDOM domain in J2. However, the modified WISSDOM used various values of wind speeds and wind directions along the sounding tracks; any number of sounding observations can be easily merged in the synthesis domain simultaneously in J6. Therefore, we have emphasized these descriptions in the revised manuscript:

L218-223. The main difference between  $J_6$  and  $J_2$  is that the sounding data with various wind speeds and directions were used as an observation for given 3D locations in  $J_6$  instead of the constraint of homogeneous background winds (i.e., uniform wind speed and direction) for each level in the studied domain in  $J_2$ . An additional benefit of  $J_6$  is that any number of sounding observations can be efficiently adopted in the WISSDOM synthesis domain

2. Different spatiotemporal resolutions of the inputs may cause problems applying the same interpretation method. More efficient ways and flexible interpolation methods are desired for the individual inputs. One of the solutions is to put the inputs into individual constraints, and then the constraints can be further applied with suitable interpretation methods. In addition, individual constraints (ex., the AWS) considered the time if the temporal resolution of inputs is equal to or higher than the time intervals of the WISSDOM outputs. We have expanded the explanations about this issue in the test as follows:

L174-182. However, various datasets with different spatiotemporal resolutions are not favorable for appropriate interpolation of given grid points of WISSDOM synthesis, and the accuracy and reliability of the background may have been significantly affected by such a variety of datasets. Thus, these different observed or model data should be treated differently to minimize uncertainties and improve accuracy. Therefore, one of the improvements in the modified WISSDOM is that these inputs were individually separated into independent constraints with flexible interpolation methods. In addition, individual constraints considered the time if the temporal resolution of the inputs was equal to or higher than the time interval of the WISSDOM outputs.

3. We have mentioned that the WISSDOM is a kind of gradient descent technique to converge toward a solution in the text in:

L135-137. The 3D winds were derived by variationally adjusted solutions to satisfy the constraints in the cost function; thus, this is a gradient decent technique to converge toward a solution.

The procedure on how to retrieve the LIDAR QVPs is still not clear to me. Could you provide a small paragraph describing the method? Do you use a VAD technique to estimate the vertical profile of horizontal wind, as suggested in Ryzhkov 2016? And if yes, how do you estimate vertical wind without having a vertical scan? Do you derive a QVP over each LIDAR or just the DGW site as suggested in Fig. 10? Please expand the text accordingly.

We now add a paragraph that describes the lidar QVP wind retrieval. We used the VAD technique. Assuming no horizontal divergence, the vertical wind can be estimated with a PPI scan with a high elevation angle. This study uses the lidar QVP retrieved from the lidar at the DGW site only.

L452-461. The QVP of horizontal and vertical winds were retrieved based on the socalled velocity-azimuth display (VAD) technique (Browning and Wexler, 1968, Gao et al., 2004). We regressed the Fourier coefficients of the Doppler velocities of the 80° PPI under the linear horizontal wind assumption and obtained the horizontal wind profile. The vertical wind was retrieved under the assumptions of constant vertical wind, zero terminal velocity of aerosol particles, and no horizontal divergence [see Kim et al. (2022) for details on the wind retrieval]. The accuracy of the retrieved wind profile is suitable for the WISSDOM wind evaluation, given the low root mean square deviation (RMSD) of < 2.5 m s-1 and high correlation coefficient of > 0.94 of horizontal wind speed as shown in the comparison against 487 rawinsondes (Kim et al., 2022).

In Fig. 16, both reviewers pointed out the artificial band resulting from the sounding, please include an explanation in the text, noting why this leads to such an artificial anomaly.

It may have been produced by the large gradient between each input when their weights were set to be the same, and the results of the WISSDOM synthesis cannot converge well. The possible explanations for this artificial band have been provided in the text.

L722-728: Notably, significant variances usually existed between the observations and reanalysis datasets due to various spatiotemporal resolutions. The results of scenario C-3 do not converge well because there was a relatively more significant gradient between each input as their weighting coefficients were set to be the same (i.e.,  $10^6$ ). In this way, the effects of poor convergences might be amplified and superposed with the AWS and lidar observations along the sounding tracks. This may be a possible reason that artificial signals existed over the DGW site in scenario C-3.

From the additions in the text, it is still not clear, why the control run is objectively the best run, especially with regard to the sensitivity tests in experiments B and C. While the control run itself is compared to measurements – albeit not independent ones – the experiments A, B and C are only compared to the control run. This does not allow us to see, whether this difference actually results in an improved or worsened performance with respect to the measurements. Hence it also does not provide the information necessary to determine that the control run performs better. If the goal is to show that the control run is optimal, then comparing all experiments to the measurements would make more sense. Since this modified version of WISSDOM includes more input data sources (i.e. more constraints in the cost function), it is valuable to reassess, whether the choice of weights (experiments C) still performs well. The parameter choice for the integration of the AWS data (experiments B) seems to be highly dependent on AWS location and topography and hence also requires a verification against measurements here. In conclusion, I would suggest to show the results of the sensitivity tests in experiments A, B and C with respect to the sounding and the QVP and adapt the discussion accordingly. With respect to the verification measurements being used in the algorithm, it is important to state clearly in the text that these are not independent measurements and that the control run is not verified independently. The further experiments A can then show, how WISSDOM performs against soundings, when soundings are not used (as an example), or against QVP, if LIDAR is not used.

Thank you very much for this valid point and valuable suggestions. Although the results of the control run provided plentiful information on the discrepancies in the studied domain for each experiment, it takes work to evaluate the difference in facts. Therefore, we have added more analysis relying on your suggestions. The discrepancies between sounding observations along its tracks and QVP above the DGW site and the results of each experiment have been well documented and discussed in this revision. Also, we have declared a statement as the control run is not verified independently with the observations at the beginning of Section 4.2, where we started the analysis in the intercomparison between the derived winds and observations. We also provided some suggestions on the setting in the WISSDOM synthesis. Please refer to the modifications:

L465-468. Because the verification observations are being used in the WISSDOM synthesis, the results of the control run are not verified independently; nevertheless, detailed discussions regarding the results of the sensitivity tests for the observations are presented in Section 5.

L636-650. In addition, the discrepancies in derived 3D winds between sounding observations and QVP were also examined along the sounding tracks (Fig. 13b) and

above the DGW site (Fig. 13c). Sounding observations played an essential role in the derived winds along its tracks. The maximum discrepancies of u- (v-) component winds are exceeded by approximately -2 (-1) m s-1 if the WISSDOM synthesis lacks sounding observations. However, small discrepancies (nearly 0 m s-1) were presented when the sounding (lidar) data were (not) implemented at all levels in A-1. The peaks in the discrepancies manifested the potential impacts from the lidar and AWS. This may be a result of lidar (AWS) having relativity higher data coverage at ~1.4 (0.8) km MSL (cf. Fig. 4). The maximum discrepancies between the derived winds and the QVP winds are approximately -4 and 4 (-1 and 0) m s-1 associated with u-, v- (w-) component winds. Generally, the results reveal similar trends in A-1~A-5, which also implies that all the inputs in the WISSDOM synthesis are equally significant against the QVP. In summary, the results of this experiment (cf. Fig. 13) show that the lidar, sounding, and AWS data are more critical inputs in modified WISSDOM. Therefore, it will be beneficial if various inputs can be included in the synthesis.

---

## Author Response (AR3)

**JGR-2022-218 Responses (highlighted in red) to Referee#1 20 Jan. 2023**

Review "High Resolution 3D Winds Derived from a Modified WISSDOM Synthesis Scheme using Multiple Doppler Lidars and Observations"

**General remarks:**

The authors took all of the major suggestions into consideration and added substantial analyses to the evaluation of the algorithm. The additional figures containing the comparison to the observational data now support the conclusions. However, I recommend to add to Figures 13, 15 and 17 b and c the control run as well. It is deemed to be the best performing setting, and this would greatly facilitate the comparison to the other experiments. If the control run is absent from the evaluation, it is still difficult to compare, whether the experiments in A, B and C performed better or worse than the control.

Moreover, the documentation and explanation of the algorithm has improved significantly. While the language has improved substantially, there are still a number of phrasing issues and the manuscript would benefit from further language editing. Some of these are highlighted in the specific remarks.

We appreciate Referee#1 reading our manuscript carefully, which help us to improve the manuscript substantially. In this round, the comparisons between the control run and observational data have been added in Figs. 13, 15, and 17b and c. We also provided the descriptions associated with the intercomparison in this revision. The language has also been corrected per your comments. A set of responses to your comments is provided below. Specific locations of modified portions (marked as underlines) were also noted as the number of lines in the revised manuscript.

Revised figures and brief descriptions related to the comparisons between the control run and observational data can be found in the following: L645-647. The discrepancies of sounding observation and control run in u- and vcomponent winds reveal relatively small values than the A-3 but similar to the other designs (purple lines in Figs. 13b).

L650-653. The QVP winds and control run discrepancies in u- and v-component winds show similar values for all designs, but relatively small values can be obtained in w-component winds (purple lines in Figs. 13c).

Figure 13. (a) Vertical profiles of averaged discrepancies of 3D winds for each design in Experiment A at 06:00 UTC on 14 Feb. 2018. The averaged discrepancies of u-, v- and w-component winds were plotted by solid, dash, and dash-dot lines, and the black, red, blue, green and orange lines indicate A-1, A-2, A-3, A-4 and A-5, respectively. (b) The same as (a) but for the discrepancies of sounding observations and u-, and v-component winds and control run (purple lines). (c) The same as (b) but for the discrepancies of QVP and w-component winds.

L702-704. Figs. 15b and 15c show the discrepancies of derived 3D winds between the sounding observations, QVP, and control run. Their patterns are similar to A-1~A5 (cf. Figs. 13b and 13c),...